# Prediction of the compressive strength of high-performance self-compacting concrete by an ultrasonic-rebound method based on a GA-BP neural network

Guoqiang Du[1], Liangtao Bu[1]*, Qi Hou[2], Jing Zhou[3], Beixin Lu[1]

1 School of Civil Engineering, Hunan University, ChangSha, Hunan Province, China, 2 Hunan Hongli Civil Engineering Inspection and Testing Co., Ltd., ChangSha, Hunan Province, China, 3 State Key Laboratory of Subtropical Building Science, South China University of Technology, Guangzhou, Guangdong Province, China

* plt63@126.com

**Data Availability Statement:** All data are available in Zenodo at http://doi.org/10.5281/zenodo. 4620872.

**Funding:** This work is supported by the National Natural Science Foundation of China(URL:http://

## Abstract

To address the problem of low accuracy and poor robustness of in situ testing of the compressive strength of high-performance self-compacting concrete (SCC), a genetic algorithm (GA)-optimized backpropagation neural network (BPNN) model was established to predict the compressive strength of SCC. Experiments based on two concrete nondestructive testing methods, i.e., ultrasonic pulse velocity and Schmidt rebound hammer, were designed and test sample data were obtained. A neural network topology with two input nodes, 19 hidden nodes, and one output node was constructed, and the initial weights and thresholds of the resulting traditional BPNN model were optimized using GA. The results showed a correlation coefficient of 0.967 between the values predicted by the established BPNN model and the test values, with an RMSE of 3.703, compared to a correlation coefficient of 0.979 between the values predicted by the GA-optimized BPNN model and the test values, with an RMSE of 2.972. The excellent agreement between the predicted and test values demonstrates the model can accurately predict the compressive strength of SCC and hence reduce the cost and time for SCC compressive strength testing.

## Introduction

Construction of large-span and super high-rise concrete structures has become a current trend [1], posing an increasingly stringent requirement for the compressive strength of concrete. For this reason, self-compacting concrete (SCC) with high strength, self-compacting ability, and excellent load-bearing performance has been gradually used in civil engineering. The existing compressive strength testing protocol generally requires that test cubes be reserved, but they are easily lost at the construction site [2]. During the project acceptance process, the compressive strength of concrete as an important acceptance criterion is a key concern of construction project participants [3]. Therefore, how to conduct the in situ testing of SCC compressive strength has become an urgent problem to be solved [4–8].

www.nsfc.gov.cn/), reference number: 51278187. The funders had no role in study design, data collection and analysis, decision to publish, or preparation of the manuscript. In addition, Q. Hou received salary from Hunan Hongli Civil Engineering Inspection and Testing Co., Ltd. The company provided great help in the design and implementation of the experiment.

Lin et al. [9] developed a backpropagation neural network (BPNN) model to predict the ultrasonic pulse velocity (UPV) of concrete using two parameters, i.e., the aggregate content and water-cement ratio of concrete. Duan et al. [10] constructed a neural network (NN) model with only one hidden layer to accurately predict the compressive strength of recycled aggregate concrete. Asteris et al. [11] established an NN model with 11 input parameters to predict the compressive strength of admixture-based concrete. Anderson and Seals [12] established prediction models for the nondestructive testing (NDT) of the compressive, flexural, and tensile strengths of six different types of concrete based on experimental results, and verified the applicability of NN models. Garzón-Roca et al. [13] combined artificial NNs (ANNs) and fuzzy logic to estimate the compressive strength of masonry structures made of clay bricks and cement mortar based on available test results. Zhou et al. [14] accurately predicted the compressive strength of hollow concrete cube masonry by combining ANNs and a fuzzy logic system. Getahun et al. [15] constructed an NN model with a single hidden layer to predict the compressive and tensile strengths of concrete incorporated with agricultural and construction wastes. Torre et al. [16] established a multilayer perceptron network to very accurately predict the compressive strength of ultra-high-performance concrete.

In the practical application of BPNN for compressive strength prediction, BPNN is susceptible to being trapped in the local optima and often has slow iterative convergence. To overcome this problem, the genetic algorithm (GA) can be used to optimize the weights and thresholds of BPNN to avoid BPNN being trapped in the local minima as well as improve the convergence rate and accuracy of BPNN [17].

In this study, a BPNN model was proposed to predict the compressive strength of SCC, and GA was used to optimize the initial weights and thresholds of the BPNN. A dataset consisting of 600 data points of UPV, rebound value, and cube compressive strength was obtained experimentally, and the compressive strength was predicted using UPV and the rebound value as input parameters. The NN was trained using the experimentally obtained dataset, and the predicted results were compared with the test results to verify the performance of the constructed NN. The model was implemented using MATLAB software. The accuracy of the model was calculated using the actual test data to demonstrate the feasibility of the model. The results showed that the proposed model can very reliably predict the compressive strength of SCC and realize the NDT of the compressive strength of SCC, to facilitate the testing of the compressive strength of SCC by construction personnel.

## Materials and methods

This study aimed to establish an ANN-based ultrasonic-rebound method for predicting the compressive strength of SCC. The UPV and rebound methods are two of the most reliable methods for nondestructive evaluation (NDE) of construction materials. In this study, the two methods were used to test SCC member specimens that were prepared using concrete grades of C50, C60, C70, C80, C90, and C100.

### Test materials

The materials used in the test included cement, river sand, limestone gravel, slag powder, fly ash, silica fume, a water reducing agent, and admixture (Fig 1). The chemical composition of raw materials is shown in Table 1. The Nanfang brand ordinary Portland cement with high activity (with a grade of 52.5 MPa) was selected for the SCC in this study. Table 2 lists the gradation of river sand; then, sand with a diameter larger than 5 mm and smaller than 160 μm was removed. The particle size of the limestone gravel aggregates was controlled at 5 to 20 mm. The property indices of S95 slag powder used in the test are listed in Table 3. To improve

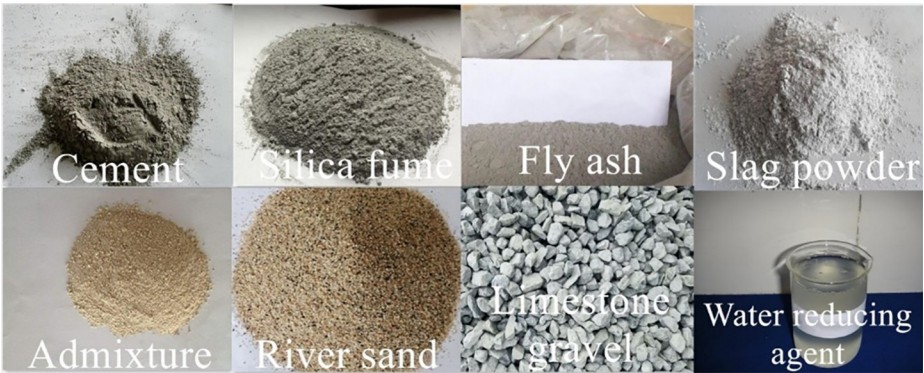

**Fig 1. Materials.**

the workability and alleviate the cracking of the concrete, grade I fly ash was used in the test; its property indices are provided in Table 4. The property indices of the microsilica fume used in the test are presented in Table 5. The water reducing agent of PCA polycarboxylic acid series was used to achieve good water reduction and ensure concrete fluidity. The ZW self-compacting nonshrinkage concrete admixture was used.

## Specimen preparation

The mix proportions of SCC using concrete grades C50 to C100 were obtained by referring to relevant literature and specifications, as shown in Table 6.

Six sets of 100 test cubes each with a size of 150 mm × 150 mm × 150 mm were prepared with concrete grades of C50, C60, C70, C80, C90, and C100 MPa to be used in the rebound, ultrasonic-rebound combined, and cube compressive strength tests [18, 19]. To prepare the test cubes, first, the mixer was started, with the drum of the mixer moistened with water to prevent the otherwise dry inner wall from affecting the water-cement ratio. Then, coarse and fine aggregates (gravel and sand) were added to the mixer and mixed well, followed by addition of the cementitious materials (fly ash, slag powder, cement, and the SCC admixture) and mixing for ten minutes until it was evenly mixed. Finally, the water reducing agent was dissolved in tap water, added to the mixer, and mixed for 15 minutes.

After mixing, the concrete was cast in the moulds and allowed to stand at room temperature for 24 hours while keeping the concrete surface moist. Then, the concrete was demoulded after its initial setting, the specimens were placed in a curing tank at a constant temperature of 60˚C for three days, removed from the curing tank, and sprinkled with water and covered with plastic film for curing at room temperature for 28 days.

## Test procedure

After 28 days of curing, the specimens were subjected to the ultrasonic, rebound, and cube compressive strength tests. The specific procedures are as follows:

**Table 1. Chemical composition of the main raw materials.**

| Raw material | Main chemical composition |
|---|---|
| Cement | CaO (64–67%), $SiO_2$ (20–23%), $Al_2O_3$ (4–8%), $Fe_2O_3$ (3–6%) |
| Silica fume | $SiO_2$ (90–95%) |
| Fly ash | $SiO_2$ (33–59%), $Al_2O_3$ (16–35%), $Fe_2O_3$ (1–19%), CaO (1–10%) |
| Slag powder | CaO (30–42%), $SiO_2$ (35–38%), $Al_2O_3$ (10–18%), MgO (5–14%) |

**Table 2. Gradation of the river sand.**

| Sieve size | 5.0 mm | 2.5 mm | 1.25 mm | 630 μm | 315 μm | 160 μm |
|---|---|---|---|---|---|---|
| Cumulative percentage retained (%) | 4 | 26 | 34 | 44 | 56 | 67 |

**UPV measurement by the ultrasonic method.** The ultrasonic method used a nonmetallic ultrasonic detector to measure the sound velocity. The cast surface of the specimen was used as a test surface, and the exact measurement point on the opposite surface of the specimen was located and coated with the coupling agent. Then, three sound velocities were measured at the ultrasonic measurement point, and their mean value was taken as the final sound velocity of the cube specimen [6, 20]. The schematic diagram of the ultrasonic method test is shown in Fig 2.

**Rebound measurement by the rebound method.** A high-strength rebound meter with an impact energy of 4.5 J was used for the test. After the sound velocity measurement was completed, the test cube coated with the coupling agent was wiped clean and then placed on a press and subjected to a compression of 30 to 50 kN (with lower compression for lower grade concrete cubes). The pair of opposite surfaces not used in the sound velocity test were each subjected to a total of eight rebound strokes. During the rebound test, the rebound hammer was perpendicular to the measurement area, the compression was applied slowly, and the reading was reset quickly. The three maximum and three minimum values were eliminated, and the mean value of the remaining ten values was used as the representative rebound value R (accurate to 0.1) of the specimen. The rebound test is schematically shown in Fig 3. Fig 4 illustrates the distribution of the ultrasonic-rebound measurement points, with the ultrasonic measurement points labelled as "1" and the rebound measurement points labelled as "2".

**Cube compressive strength test.** After the ultrasonic and rebound tests, each test cube was directly compressed to failure (Fig 5) at a compression rate of 8–10 kN/s to obtain the compressive strength $f_{cu}^c$ of the test cube (accurate to 0.1 MPa [21]).

The detailed laboratory protocols has been deposited in protocols. io(dx.doi.org/10.17504/protocols.io.btjnnkme [PROTOCOL DOI]).

## Test results

Each cubic specimen was tested successively using the UPV method, the rebound method, and the cube compressive strength test. The test data were recorded and compiled into datasets for the rebound value, UPV, and cube compressive strength. The rebound values and UPVs increased with the compressive strength grade of the test cubes. These data were divided into three groups according to R, $V_P$ and $f_c$, and used to produce the three different boxplots shown in Figs 6–8 below.

## Comparative analysis

The rebound and ultrasonic methods are the two most commonly used methods for NDT of the compressive strength of concrete. The rebound value is affected by factors such as the test specimen's surface smoothness, size, shape, hardness, surface and internal concrete moisture, and cement type. The rebound method is convenient and quick and can be used on a single

**Table 3. Property indices of the slag powder.**

| Material | Strength | Density (g/cm³) | Specific surface area (m²/kg) | Loss on ignition (%) | Chloride ion (%) | Fluidity ratio (%) | Water content (%) |
|---|---|---|---|---|---|---|---|
| Slag powder | S95 | 2.89 | 425 | 0.60 | 0.036 | 102 | 0.28 |

**Table 4. Property indices of the fly ash.**

| Material | Grade | Fineness (%) | Water demand ratio (%) | Loss on ignition (%) | Water content (%) |
|----------|-------|--------------|------------------------|----------------------|-------------------|
| Fly ash | I | 12 | 92 | 3.8 | 0.1 |

concrete surface. The ultrasonic test method requires measurement on both sides of a concrete element, and it is the most popular technique for testing the compressive strength of concrete. The ultrasonic method is suitable for evaluating the uniformity of concrete. There are many factors that affect the UPV, but they do not necessarily affect the compressive strength of concrete, thus making it very difficult to directly evaluate the concrete compressive strength using this method. To date, the theoretical relationship between the rebound value, the UPV, and the concrete compressive strength has been proposed by many researchers. Table 7 lists the relational expressions for the two internationally recognized NDT methods, i.e., the UPV measurement, the rebound measurement, and their combination. Fig 9 compares the equations for UPV measurement and Fig 10 compares the equations used to measure the rebound value. Fig 11 compares the equations for measurement made by combining the two methods; the images show that the concrete compressive strengths calculated using these relational expressions differ considerably and therefore need further improvement. In this study, ANN was used to predict the concrete compressive strength.

## Strength prediction model

### ANN model development

An ANN is a computational model that simulates the biological neural structure. It is composed of many interconnected neurons, with each node representing an output function (excitation function) [31]. The weights that connect each neuron represent the effects of input parameters on the output of the neuron and can be adjusted to produce a desired output. ANN is designed to learn from the existing data, which are transferred from the input layer to the output layer while the deviation between the actual value and the output value is minimized, thereby achieving the mapping of input parameters to a given output [11, 32, 33]. The NN architecture is shown in Fig 12. Each neuron receives the weighted input from the neuron in the previous layer as the input, which is then transferred to other neurons through the activation function, so the information is represented by many cross-connected weights [34]. The ultimate goal is to minimize the error between the actual output and the expected output.

The backpropagation (BP) algorithm is commonly employed to optimize parameters in NN algorithms and BPNN has been widely adopted in civil engineering applications [35, 36]. Using the BP algorithm, the signal experiences both forward propagation and backward error propagation. In forward propagation, the signal is transmitted from the input layer through the hidden neurons to the output layer. In BP, the output error is calculated backwards according to the original path through the hidden layer from the output layer. In this process, the weights and thresholds are continuously updated until the output error of the network reaches an acceptable level [37–40]. Traditionally, the BP algorithm determines the weights in the network by the gradient descent method, which has a low computational speed due to its linear

**Table 5. Property indices of the silica fume.**

| Material | Loss on ignition (%) | Chloride ion (%) | Silica (%) | Specific surface area (m$^2$/kg) | Water content (%) | Water demand ratio (%) | Activity index |
|----------|----------------------|------------------|------------|-----------------------------------|-------------------|------------------------|----------------|
| Silica fume | 2.5 | 0.014 | 94.05 | $2.51 \times 10^4$ | 1.1 | 113 | 112 |

**Table 6. Mix proportions of SCC.**

| Test ID | Cementitious materials | | | | | | Water (kg) | Sand (kg) | Gravel (kg) | Water reducing agent (kg) |
|---|---|---|---|---|---|---|---|---|---|---|
| | Total (kg) | Cement (kg) | Silica fume (kg) | Fly ash (kg) | Slag powder (kg) | Admixture (kg) | | | | |
| A | 331.21 | 226.86 | 0.00 | 13.61 | 56.71 | 34.03 | 104.06 | 446.70 | 550.00 | 0.95 |
| B | 353.10 | 254.38 | 15.10 | 15.26 | 30.20 | 38.16 | 97.24 | 440.81 | 550.00 | 1.01 |
| C | 375.32 | 251.32 | 33.74 | 18.82 | 33.74 | 37.70 | 93.01 | 402.58 | 550.00 | 2.15 |
| D | 376.94 | 253.07 | 33.51 | 18.89 | 33.51 | 37.96 | 86.21 | 411.38 | 550.00 | 2.51 |
| E | 359.91 | 226.87 | 44.11 | 21.10 | 33.80 | 34.03 | 79.12 | 417.07 | 550.00 | 2.75 |
| F | 347.82 | 224.00 | 44.66 | 11.27 | 34.30 | 33.60 | 74.73 | 420.00 | 550.00 | 2.99 |

convergence. The Levenberg-Marquardt algorithm has been adopted to increase the speed due to its use of approximate second derivatives.

## Information transfer of BPNN

BPNN is a feedforward multilayer ANN. The process of information transfer through a single neuron in the hidden layer of the BPNN is shown in Fig 13. The BPNN can be described as follows:

$$I-H_1-H_2-H_3-\ldots H_n-O \tag{1}$$

where I is the number of input neurons, $H_n$ is the number of hidden neurons in the nth layer, n is the number of layers of hidden-layer neurons, and O is the number of output neurons.

The choice of the activation function f has a considerable influence on the performance of an NN model. The sigmoid function is the most commonly used activation function, but it is important to choose the most appropriate activation function for different research objects. Reference [41] presents a thorough description of a large number of transfer functions. In the present study, the default transfer functions of BPNN (i.e., the tansig function for the hidden layer and the Purelin function for the output layer) are adopted.

## Test data

An appropriate dataset is necessary to train a reliable NN model. The dataset must be obtained through actual experiments and cover all possible data ranges. Because the NN model is developed through training with an existing dataset, the NN would not produce accurate prediction

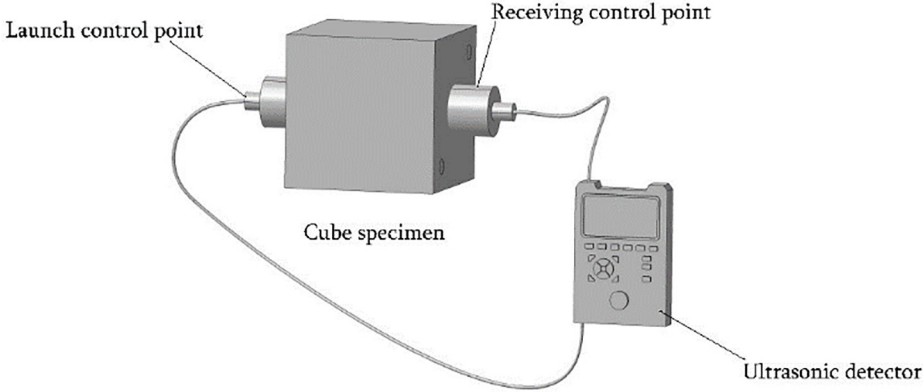

**Fig 2. Schematic diagram of the ultrasonic test.**

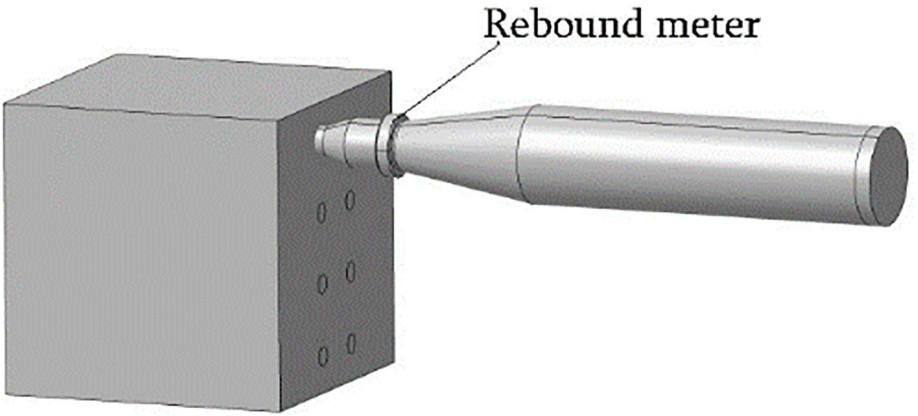

**Fig 3. Schematic diagram of the rebound test.**

results if the data in the dataset are inaccurate. In this study, the dataset was obtained through rigorous testing, where the rebound value, UPV, and cube compressive strength of each concrete specimen were measured successively. Fig 14 shows the frequency histograms of the data. Table 8 presents the mean, maximum, and minimum values of the test data, as well as the standard deviations of R, $V_P$, and $F_C$ for each group of test cubes.

The database has the following advantages:

1. The database provides a sufficient number of test data, that is, the experimentally measured rebound values, UPVs, and cube compressive strengths of 600 specimens.

2. The test data were measured under the same test conditions, by the same person and with the same equipment to eliminate measurement errors from equipment differences.

3. The data cover most possible cases. In Table 8, the rebound values range from 46.94 to 75.10, and the UPV values vary between 3.86 and 4.60 km/s, which are suitable for SCC with a compressive strength in the 41.52 to 102.1 MPa range.

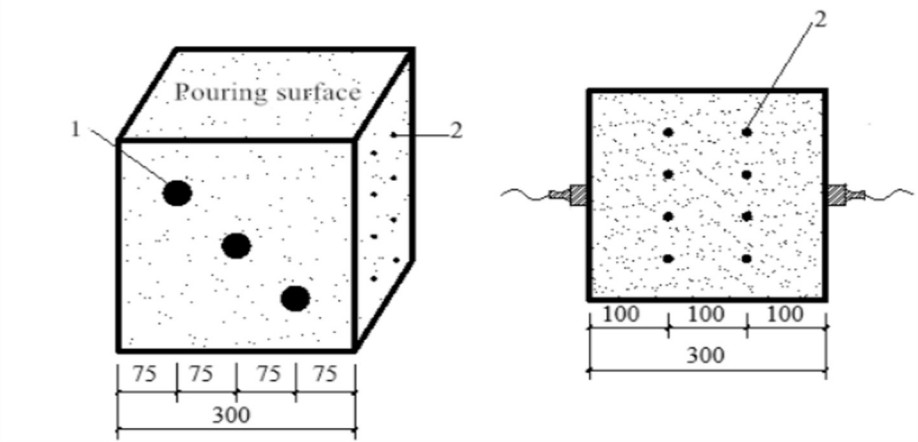

**Fig 4. Distribution of ultrasonic and rebound measurement points.**

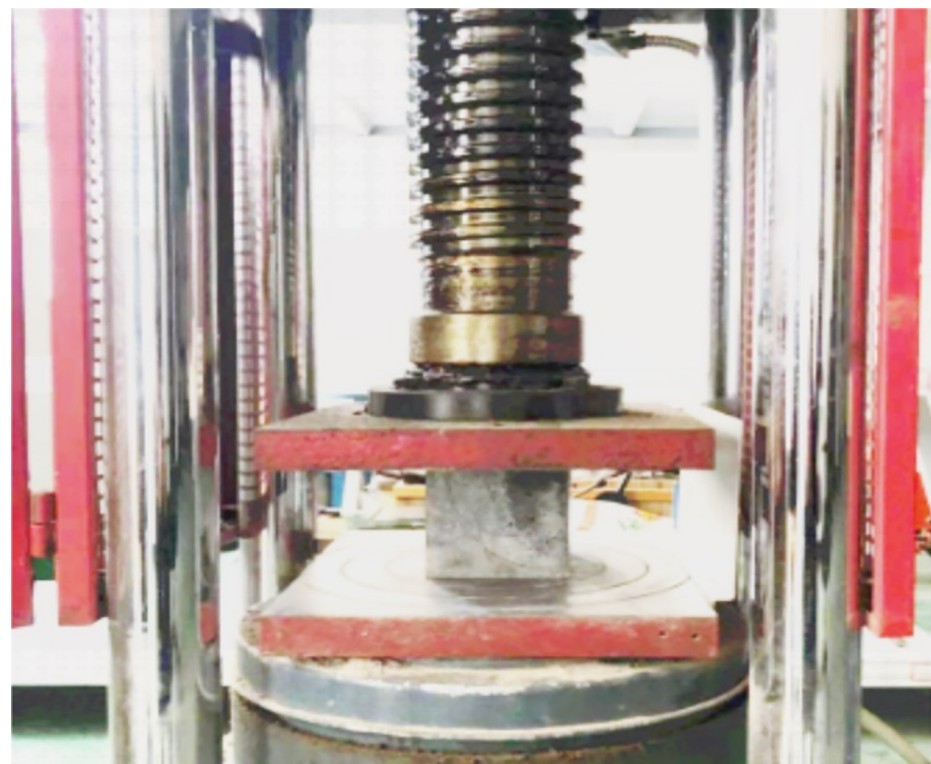

**Fig 5. Cube compressive strength test.**

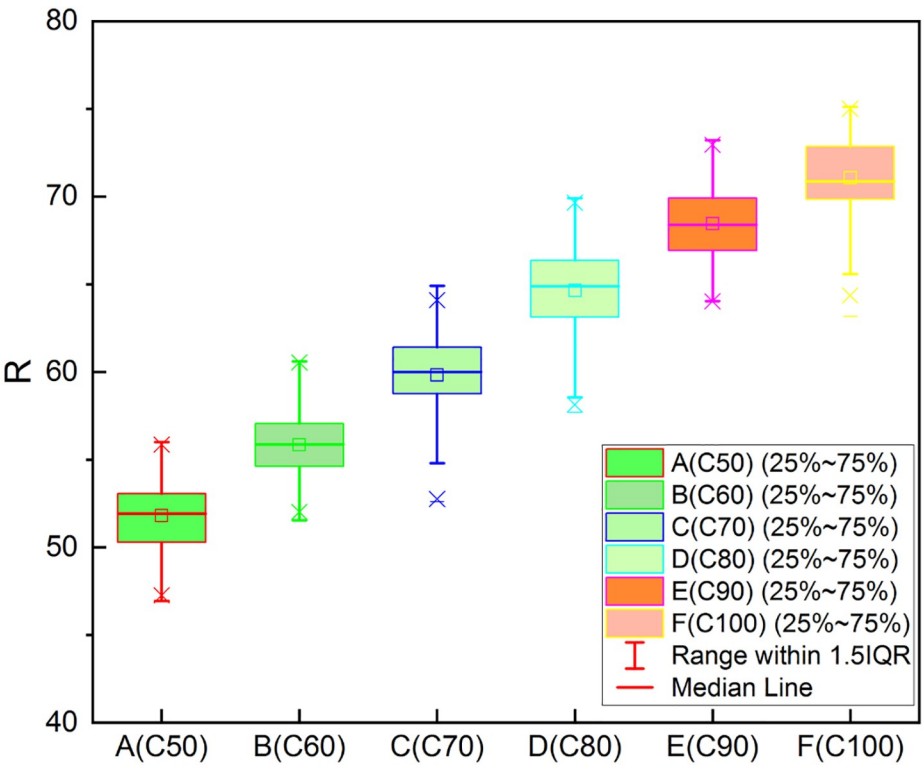

**Fig 6. Boxplot of experimental data (R).**

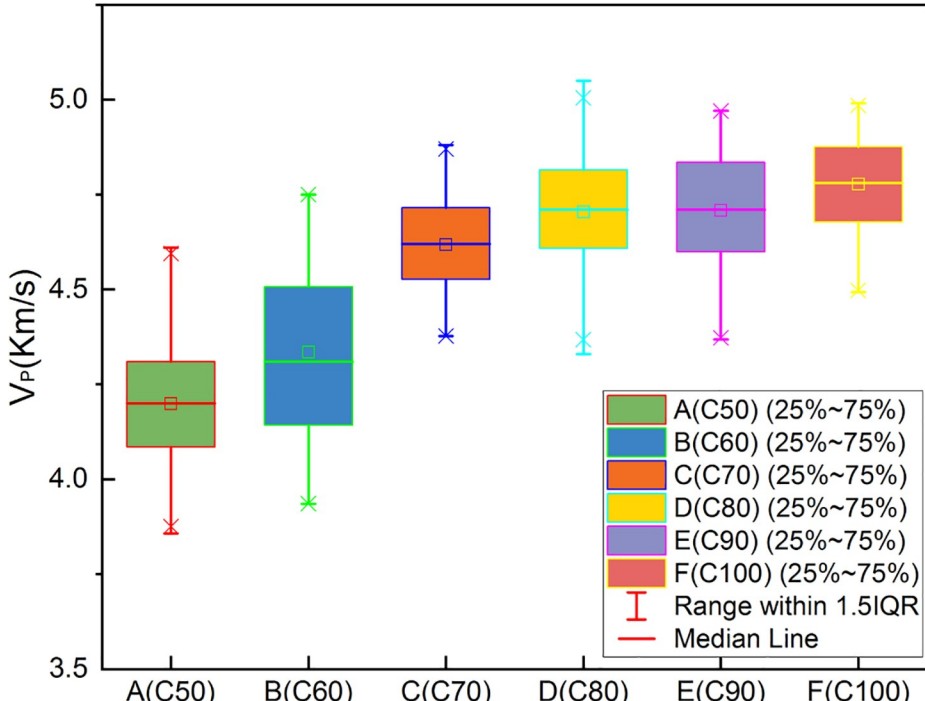

**Fig 7. Boxplot of experimental data (V$_P$).**

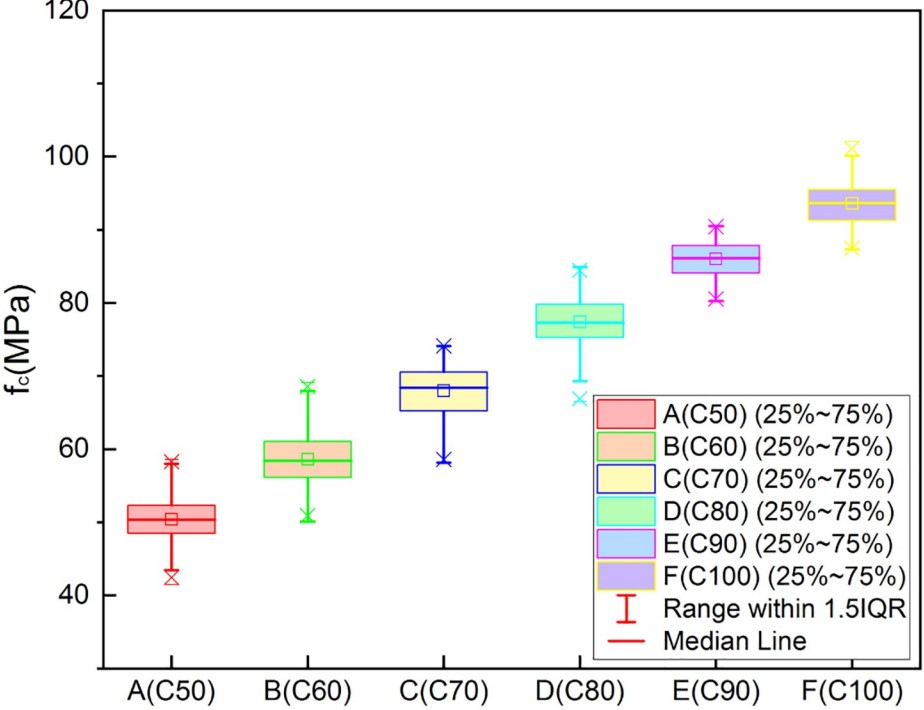

**Fig 8. Boxplot of experimental data (fc).**

**Table 7. Empirical relational expressions for estimating the compressive strength of concrete.**

| Equation | ID | Reference |
|---|---|---|
| $f_c(V_p) = 1.146e^{0.77V_p}$ | E1 | Turgut [22] |
| $f_c(V_p) = 176.9 - 96.467V_p + 13.906(V_p)^2$ | E2 | Logothetis [23] |
| $f_c(V_p) = 0.085e^{1.288V_p}$ | E3 | Trtnik et al. [24] |
| $f_c(V_p) = 1.19e^{0.715V_p}$ | E4 | Nash't et al. [25] |
| $f_c(V_p) = 8.4*10^{-9}*(V_p*10^3)^{2.5921}$ | E5 | Kheder [26] |
| $f_c(V_p) = 1.2*10^{-5}*(V_p*10^3)1.7447$ | E6 | Kheder [26] |
| $f_c(R) = -9.40 + 0.52R + 0.02R^2$ | E7 | Logothetis [23] |
| $f_c(R) = 0.4030R^{1.2083}$ | E8 | Kheder [26] |
| $f_c(R) = 1.353R - 17.393$ | E9 | Qasrawi [27] |
| $f_c(V_p, R) = e^{1.78\ln(V_p)+0.85\ln(R)-0.02} * 0.0981$ | E10 | Logothetis [23] |
| $f_c(V_p, R) = 18.6 * e^{0.515V_p+0.019R} * 0.0981$ | E11 | Arioglu and Manzak [28] |
| $f_c(V_p, R) = (0.10983 + 0.00157R - 0.79315(V_p/10) - 0.00002R^2 + 1.29261(V_p/10)^2) * 10^3$ | E12 | Amini et al. [29] |
| $f_c(V_p, R) = 0.42R + 13.166V_p - 40.255$ | E13 | Erdal [30] |
| $f_c(V_p, R) = 0.0158(1000V_p)^{0.4254} * R^{1.1171}$ | E14 | Kheder [26] |

## Data standardization

Data standardization is a critical step in the NN technology. To avoid the problem of a low learning rate in NN models, the values of the corresponding parameters of data standardization should be within the corresponding ranges. In this study, the input and output parameters

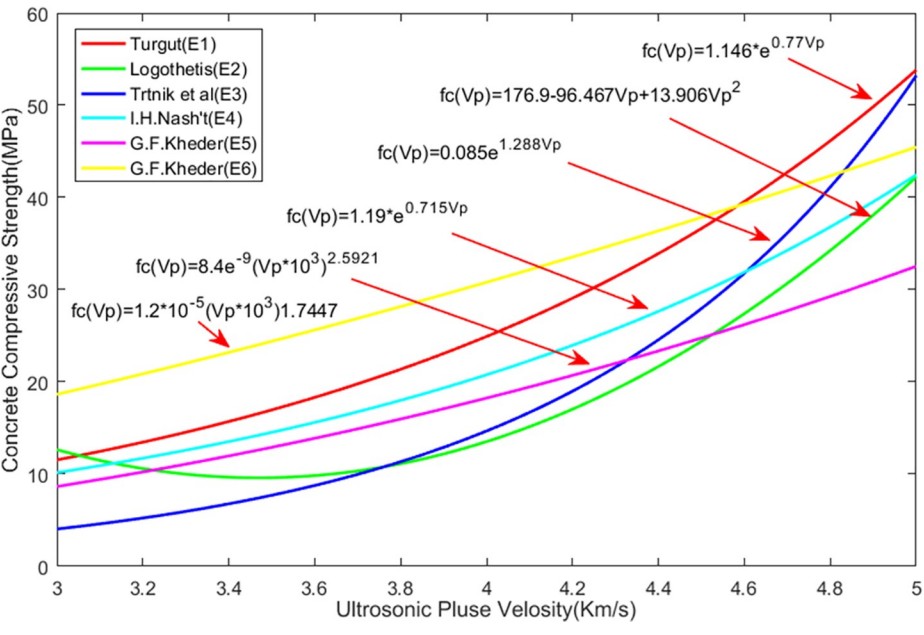

**Fig 9. Calculation equations of UPV and compressive strength.**

are normalized in the range of [–1, 1]. The data normalization equation is as follows:

$$y_i = 2 * \frac{y - y_{\min}}{y_{\max} - y_{\min}} - 1 \tag{2}$$

where $y_i$ is the normalized data, $y$ is the raw data, and $y_{\max}$ and $y_{\min}$ are the maximum and minimum values of the original data, respectively.

## Performance of the model

The parameters need to be properly selected to construct the best prediction model. Therefore, it is necessary to measure the goodness of fit of the prediction model using indices such as the Pearson correlation coefficient (P), mean square error (MSE), mean absolute error (MAE), and absolute percentage error (APE), etc. In this study, P and RMSE are used to evaluate the model performance. The higher P is, the better the fit between the experimental and predicted values is. The lower the RMSE is, the more accurate the prediction result is. The calculation equations follow.

The Pearson correlation coefficient, also known as the simple correlation coefficient, describes the linear correlation between two variables. The Pearson coefficient is commonly represented by P, which is given in Eq (3).

$$P = 1 - \frac{\sum\limits_{i=1}^{N} (X_i - Y_i)^2}{\sum\limits_{1}^{N} (X_i - \bar{X})^2} \tag{3}$$

The MSE reflects the degree of difference between the predicted and expected values. It is calculated according to Eq (4).

$$MSE = \frac{\sum\limits_{i=1}^{N} (X_i - Y_i)^2}{N} \tag{4}$$

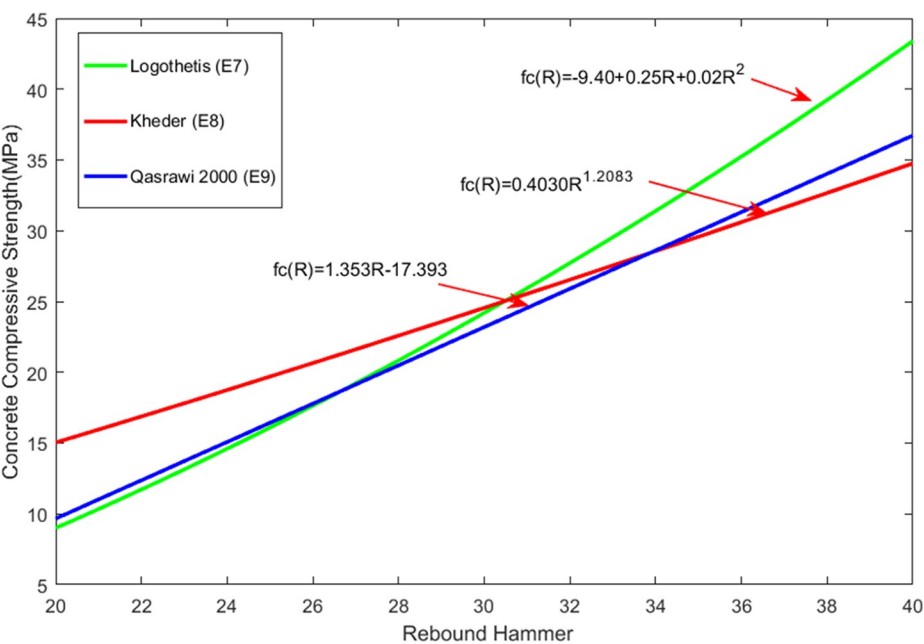

**Fig 10. Calculation equations of rebound value and compressive strength.**

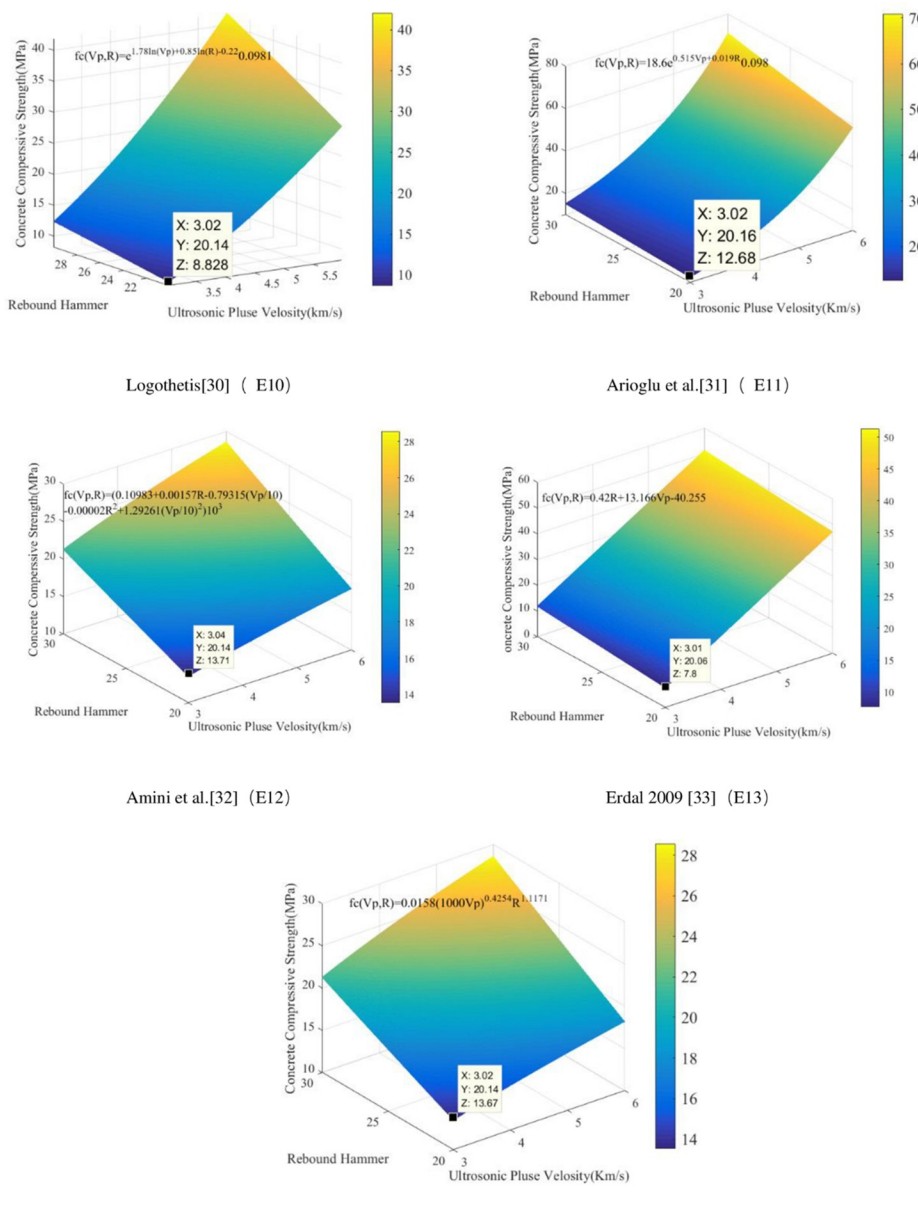

**Fig 11. Images of empirical relationship functions for compressive strength using combined measurements.**

The root mean square error (RMSE) is calculated by Eq (5).

$$\text{RMSE} = \sqrt{\text{MSE}} \tag{5}$$

where $X_i$ is the expected output of the ith sample, $\bar{X}$ is the sample mean, $Y_i$ is the predicted output of the model, and N is the number of samples.

## GA-optimized NN

A GA is a parallel stochastic search optimization method that simulates the theory of genetic and biological evolution in nature. Similar to the biological evolution principle of "survival of

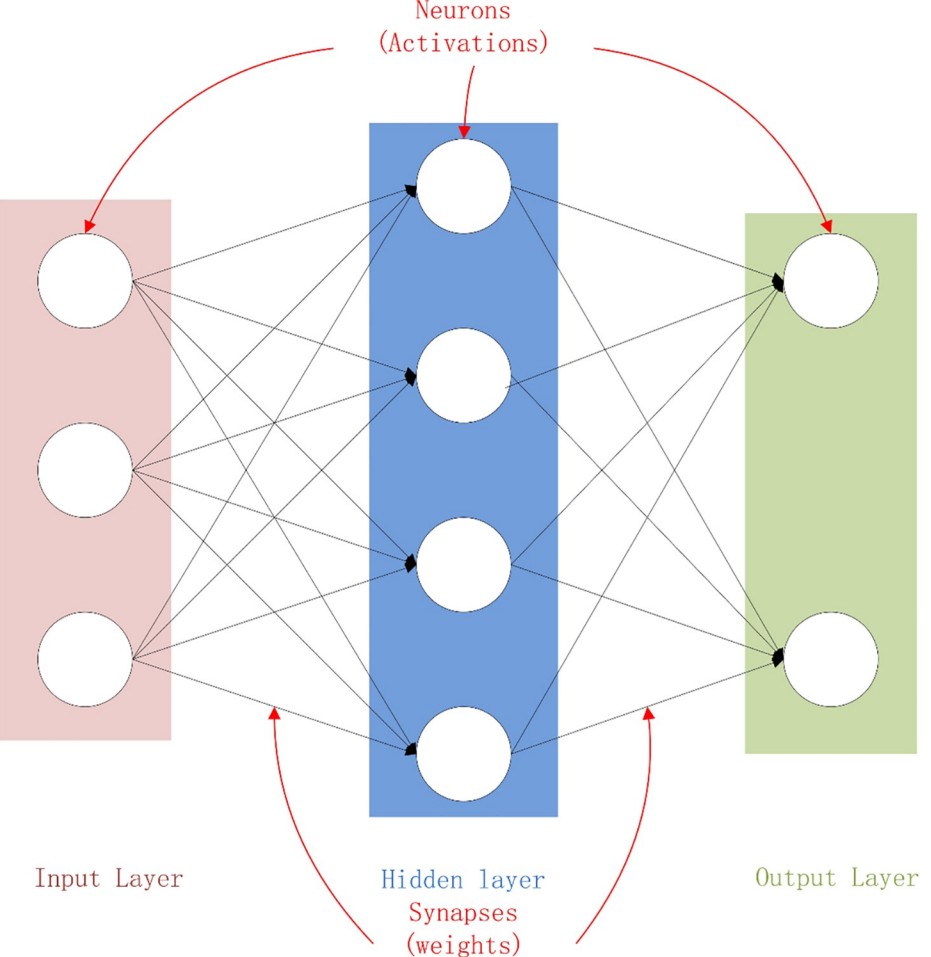

**Fig 12. Diagram of the NN architecture.**

the fittest" in nature, a GA creates an encoded tandem population by introducing optimization parameters and ranks individuals in the population according to the chosen fitness function through genetic operations (i.e., selection, crossover, and mutation); thus, individuals with higher fitness values are retained while those with low fitness values are eliminated. The new offspring population not only inherits the information from its parent generation but also outperforms its parent generation. The generational iteration continues until the stopping criterion is met. The flowchart of the GA-optimized BPNN algorithm is shown in Fig 15.

## Results and discussion

### Development of ANN model

A total of 57 different BPNN models were developed in this study. For each model, the 600 experimentally obtained data points were randomly divided into three parts: 420 data points (70%) for training, 90 data points (15%) for verification, and 90 data points (15%) for testing. A neural network model with only one hidden layer can reliably perform any prediction task [42]. The number of neurons is usually determined using an empirical formula or by trial and error. Therefore, the neural network is set to have one hidden layer containing 2 to 20 neurons

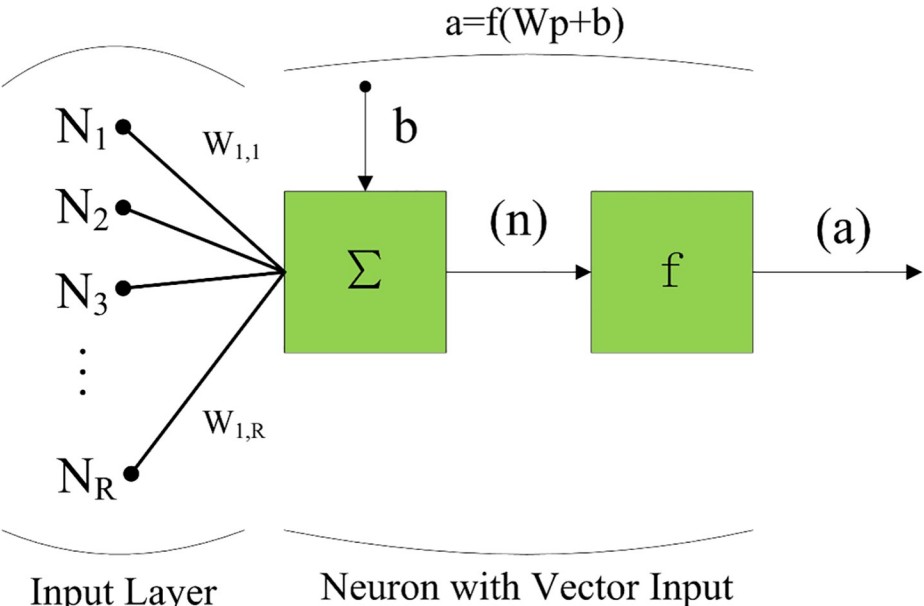

**Fig 13. Information transfer through a single neuron in BPNN.**

[43–47]. The transfer functions consist of a hyperbolic tangent sigmoid transfer function in the hidden layer and a linear purelin transfer function in the output layer. The MSE is used as a criterion for terminating the neural network training. A lower MSE reflects more ideal

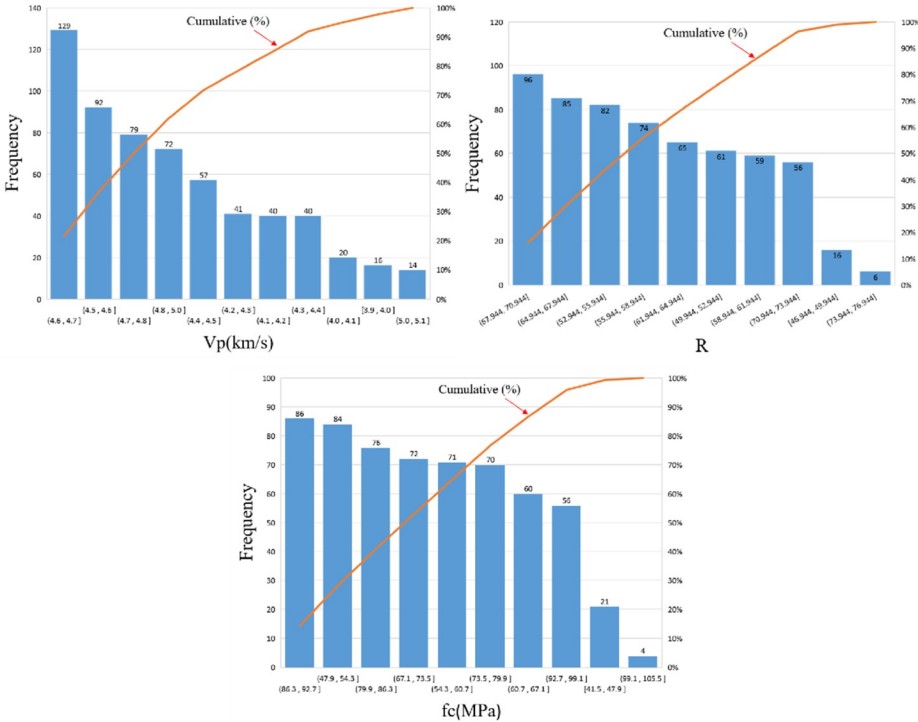

**Fig 14. Frequency histograms of the data.**

**Table 8. Statistics of the test datasets.**

| Test data | Unit | Data type | Minimum | Mean | Maximum | Standard deviation | | | | | |
|---|---|---|---|---|---|---|---|---|---|---|---|
| | | | | | | C50 | C60 | C70 | C80 | C90 | C100 |
| Rebound value (R) | - | Input | 46.94 | 61.90 | 75.10 | 1.87 | 0.17 | 3.17 | 1.87 | 0.17 | 3.17 |
| UPV ($V_p$) | km/s | Input | 3.86 | 4.60 | 5.05 | 1.65 | 0.22 | 3.79 | 1.65 | 0.22 | 3.79 |
| Compressive strength ($f_c$) | Mpa | Output | 41.52 | 72.30 | 102.1 | 2.25 | 0.13 | 3.74 | 2.25 | 0.13 | 3.74 |

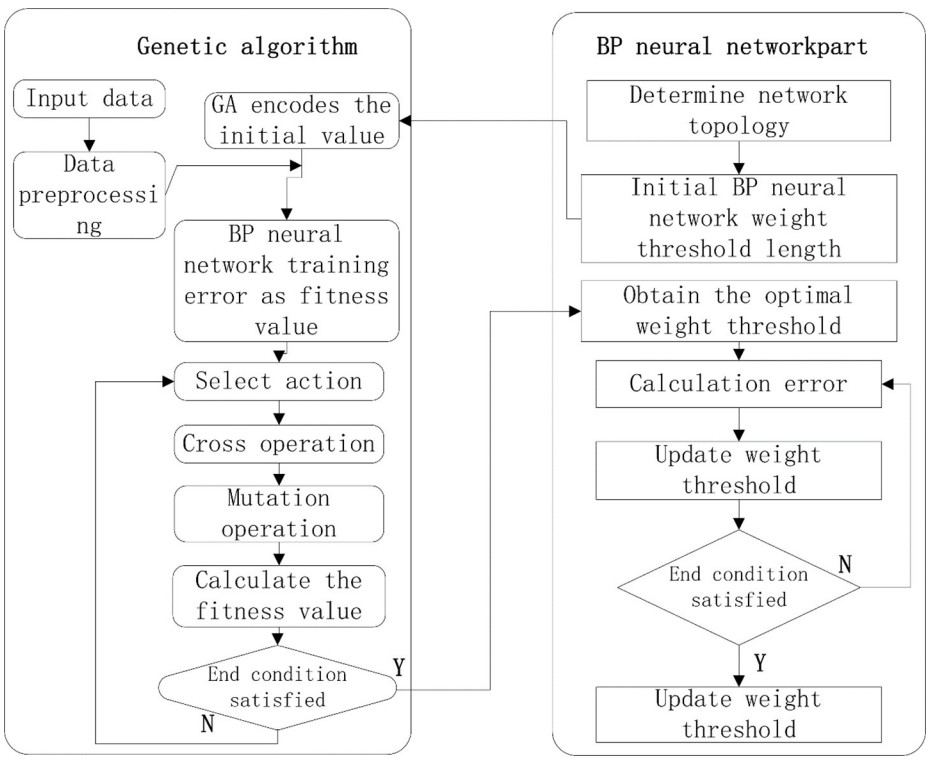

**Fig 15. Flowchart of the GA-optimized BPNN algorithm.**

**Table 9. BPNN parameters.**

| Parameters | Set value |
|---|---|
| Training algorithm | Levenberg-Marquardt algorithm |
| Number of hidden layers | 1 |
| Number of neurons in a single hidden layer | 2–20 |
| Standardization | [−1,1] |
| Network performance | RMSE, P |
| Activation function | Sigmoid, Purelin |

network performance. The correlation coefficient P is used to measure the correlation between the output and the target in the network, and the RMSE is used to evaluate the performance of the generated network. The BPNN parameter settings are shown in Table 9.

## Determination of the ANN architecture

To determine the optimal NN model for SCC compressive strength prediction, 57 different BPNN models and three NN structures with different input parameters were developed, as shown in Table 10.

The developed NN model was selected based on the RMSE values, and the results of the three optimal structures are shown in Table 11. Fig 16 shows how the number of hidden-layer neurons affects the performances of the three different BPNN architectures.

**Table 10. BPNN structure based on different input parameters.**

| Case | Input parameters | Number of input parameters |
|------|------------------|----------------------------|
| 1 | $V_P$ | 1 |
| 2 | R | 1 |
| 3 | $V_P$, R | 2 |

**Table 11. Statistical index of different optimal BPNN structures.**

| Case | Optimum BPNN model | R | RMSE |
|------|--------------------|---|------|
| 1 | 1-13-1 | 0.935 | 3.995 |
| 2 | 1-12-1 | 0.912 | 4.262 |
| 3 | 2-19-1 | 0.967 | 3.703 |

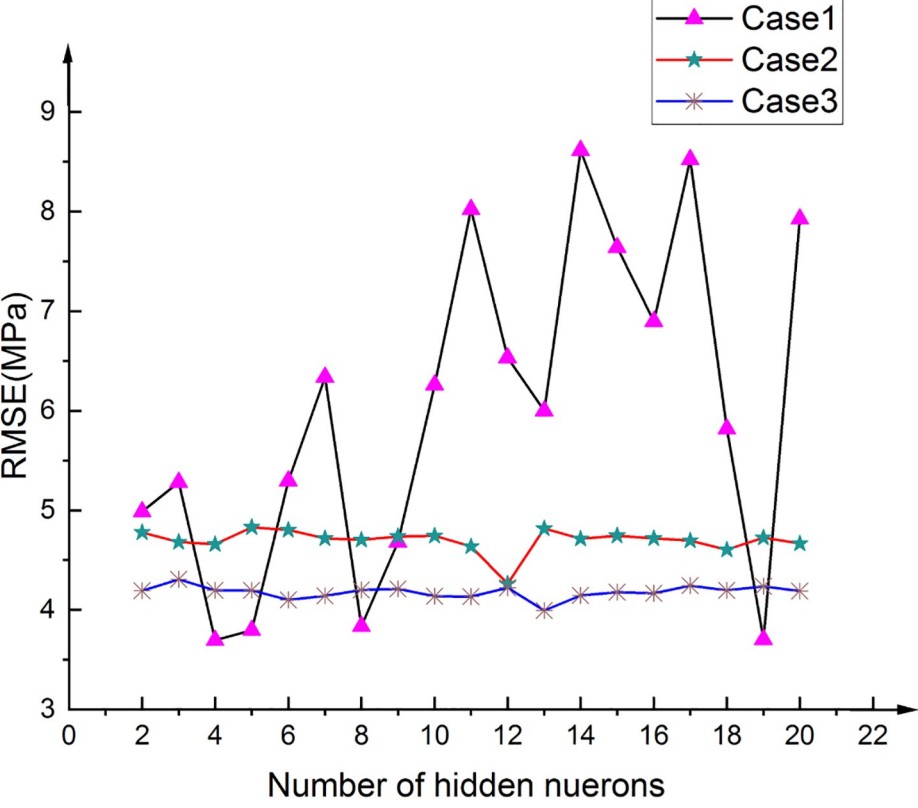

**Fig 16. Variation in the RMSE with the number of hidden neurons.**

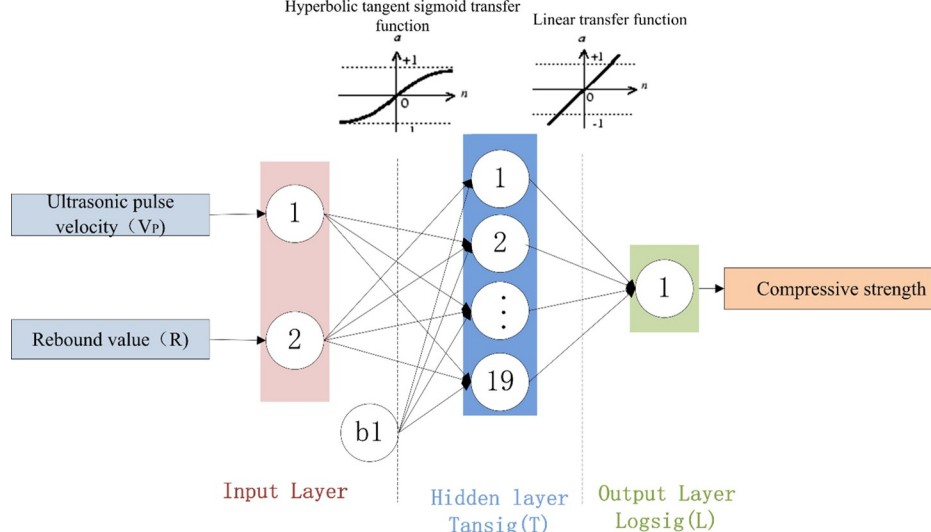

**Fig 17. Optimal BPNN structure with two input parameters.**

An examination of the data presented in Table 11 shows that 2-19-1 is the optimal BPNN. The structures of the three BPNN models are shown in Figs 17–19. It is useful to develop three optimal BPNN models because only $V_P$ or R can sometimes be measured in practice.

## Optimization by GA

The optimal structure (2-19-1) of the BPNN model was determined experimentally. The BPNN output results before optimization by GA are shown in Fig 20.

Fig 21 shows the performance of the best BPNN in terms of the MSE of the network, depicting the gradual decrease in errors as the NN was trained on the specified training set to perform learning. The figure consists of three lines. The blue line represents the gradually

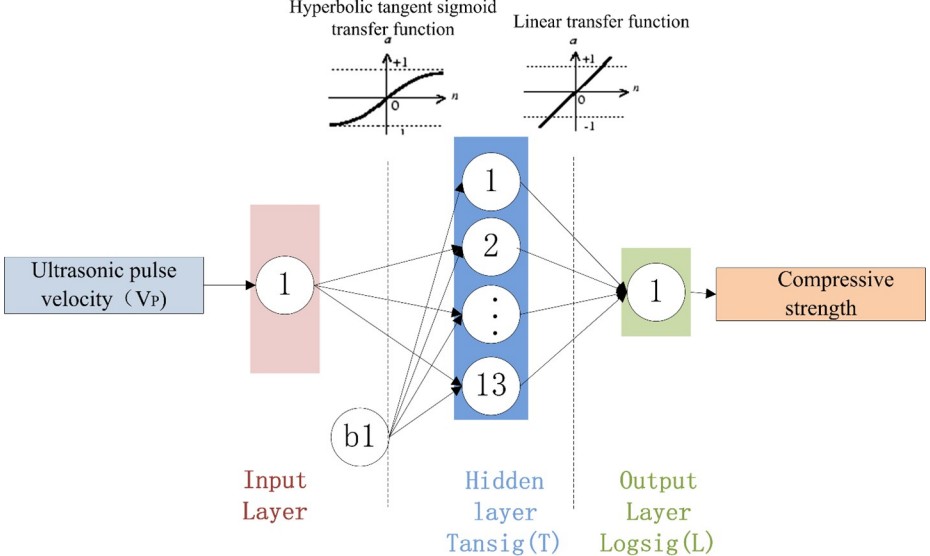

**Fig 18. Optimal BPNN structure with one input parameter ($V_P$).**

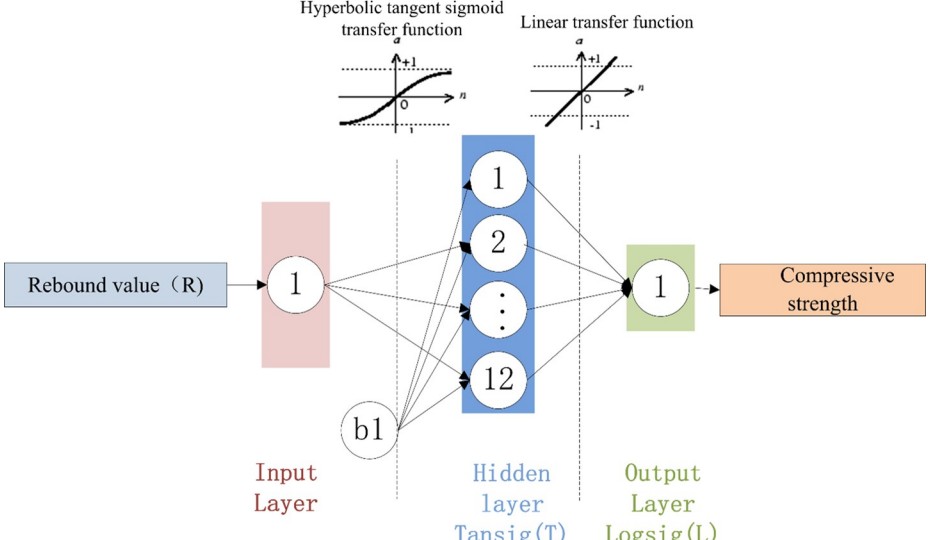

**Fig 19. Optimal BPNN structure with one input parameter (R).**

decreasing error on the training data and the green line shows the validation error. The training stopped when the validation error no longer decreased, which essentially avoided the problem of overfitting. The prediction error on the training set demonstrated the fit of our model, while the error based on the validation set measured the performance of the model in predicting new data. The red line exhibits the error on the test data, showing the generalization of the data by the model. Fig 22 presents the training state of the network.

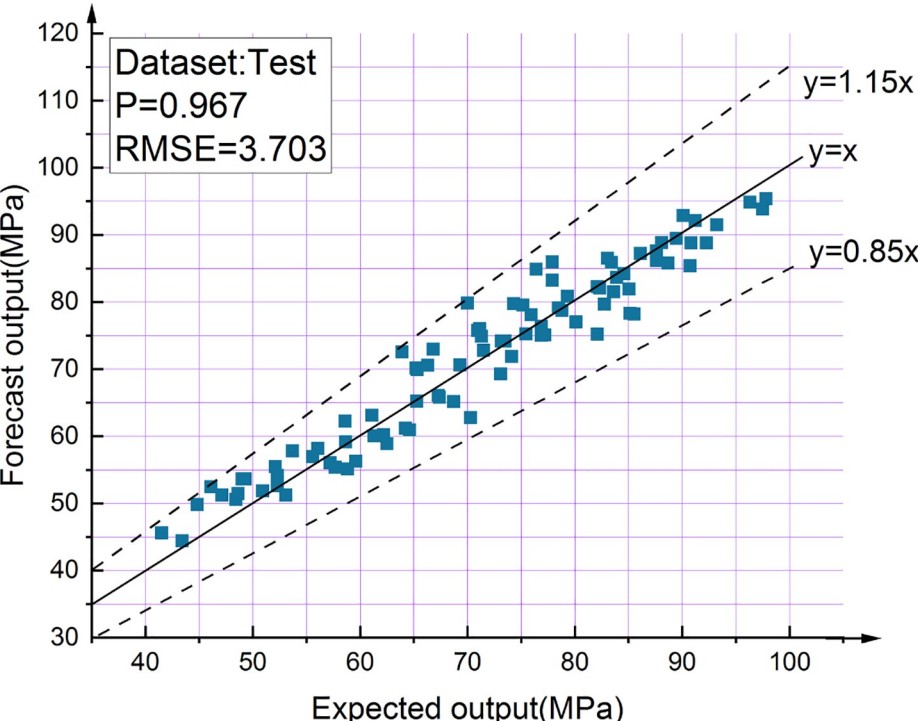

**Fig 20. BPNN output results before optimization.**

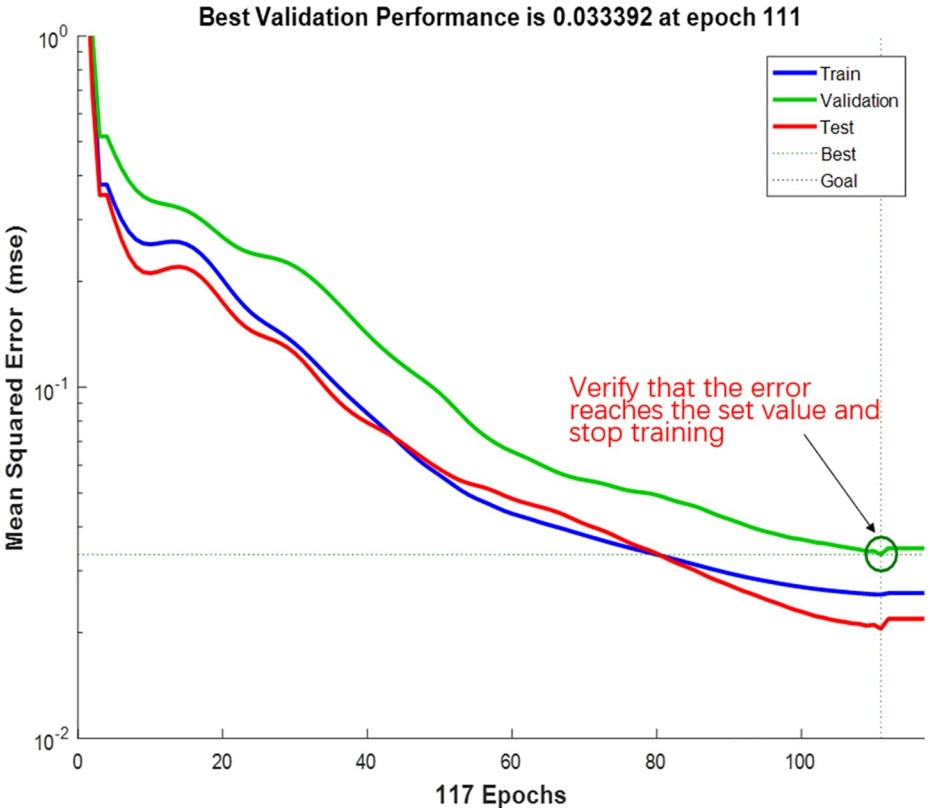

**Fig 21. Performance of the model.**

The GA transforms the decision parameters of an optimization problem into chromosomes using encoding methods and converts the optimization objective function into a fitness function that serves as the basis for evaluating the merits of the chromosomes and genetic operations [48–50]. In the present study, BPNN was organically combined with GA to improve the accuracy of the NN model. The optimization of BPNN by GA was divided into the three parts of determination of BPNN architecture, optimization by GA, and prediction by BPNN. The parameters optimized by GA were the initial weights and thresholds of the BPNN. The individual fitness values were calculated using the fitness function, and GA searched for the individual corresponding to the optimal fitness value via selection, crossover, and mutation operations. The initial weights and thresholds of the NN are generally random numbers initialized to the interval [-0.5, 0.5], and they significantly influence the NN performance. For this reason, GA was introduced to find the optimal initial weights and thresholds.

The output result of BPNN optimized by GA is shown in Fig 23. The results show that the accuracy of the BPNN optimized by GA is much better than that before. Only one test data's prediction deviation is more than 10%, and the other data's deviation is within 10% (the points between the two dashed lines in Fig 23), The correlation coefficient between the test value and the predicted value is 0.979. Fig 24 shows excellent agreement between the test results of 90 test samples and the prediction results of the best model.

## Comparisons

All the experimental data are predicted using the proposed 2-19-1 BPNN model and the 14 empirical formulas presenting in Table 7. Table 12 ranks the methods according to the RMSEs

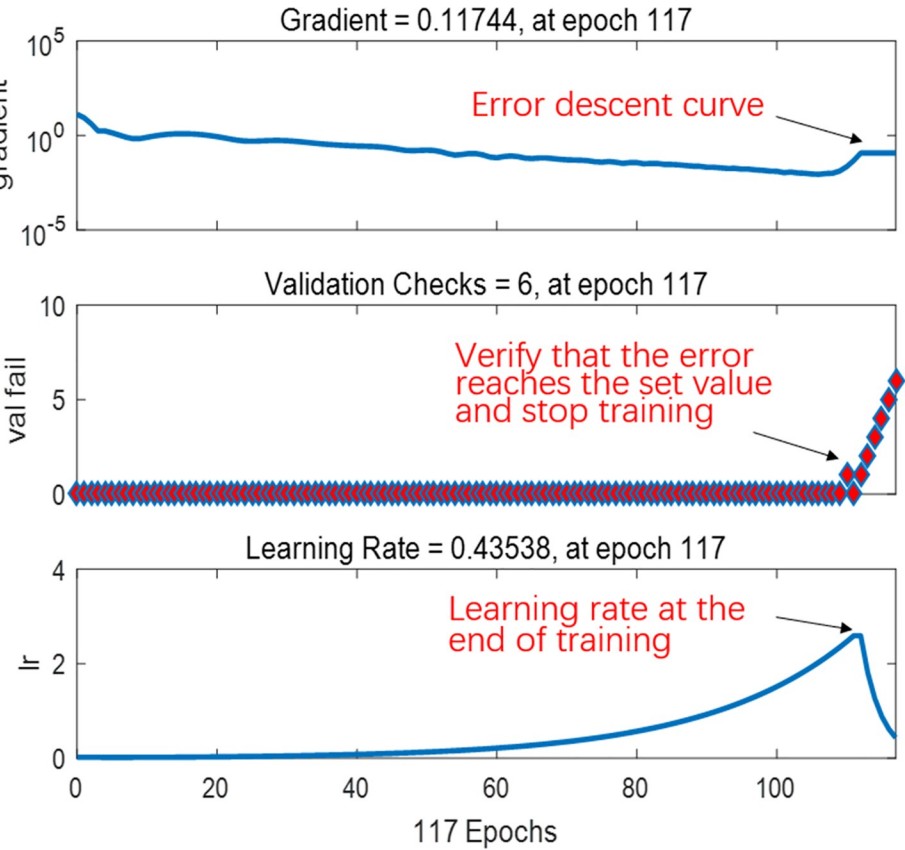

**Fig 22. Training state of the network.**

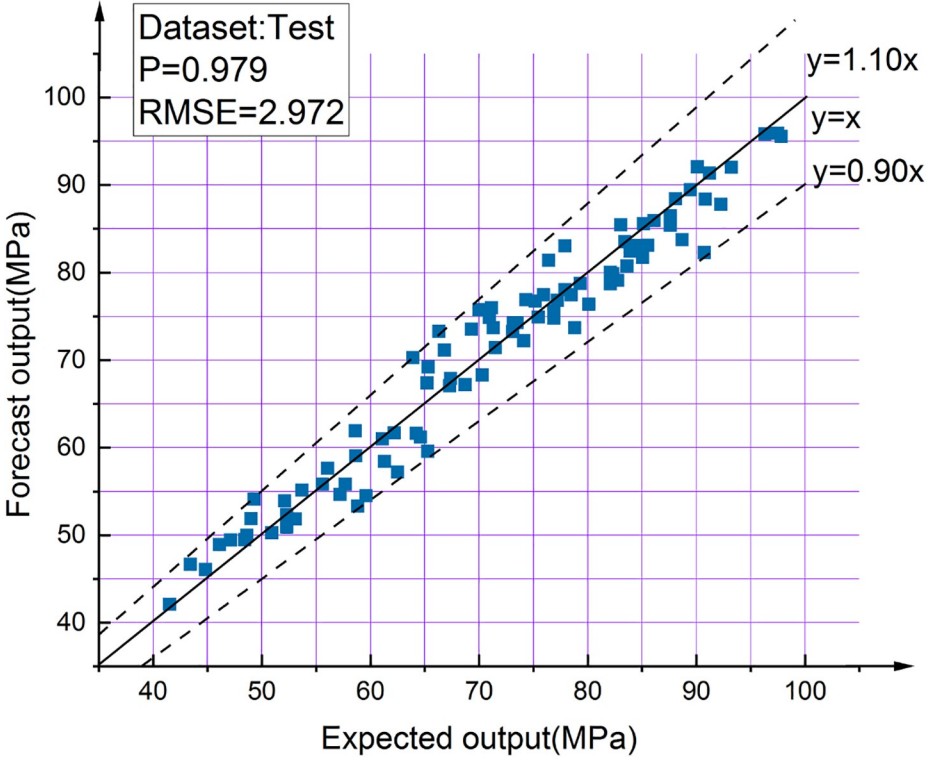

**Fig 23. GA-BPNN output results.**

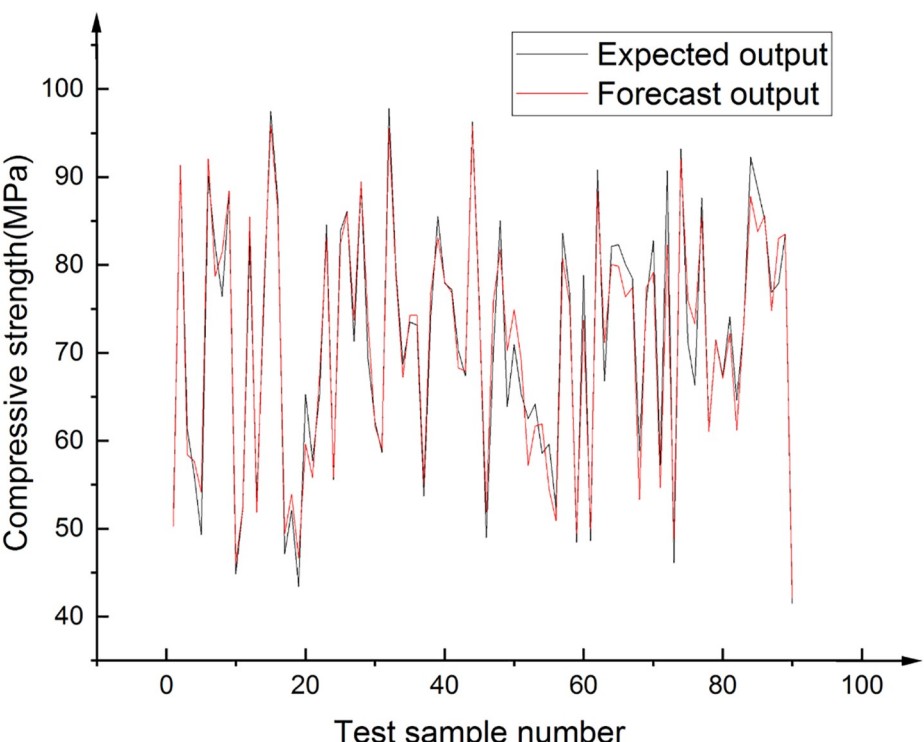

**Fig 24. Comparison of test data and predictions by the best model.**

of the predicted results. Fig 25 shows the predictions of the first six models, the prediction results of other models are shown in S1 Fig. The predictions of the proposed GA-BPNN model are closest to the experimental results and hence more reliable. The previously developed empirical formulas fail to eliminate the influences of the differences in the concrete materials and mix proportions on the compressive strength and thereby produce large prediction errors.

**Table 12. Models for concrete compressive strength prediction ranked according to RMSEs.**

| Rank | Mathematical model | Parameters | References | P | RMSE |
|------|--------------------|------------|------------|------|------|
| 1 | 2-19-1 | $V_P$, R | - | 0.9761 | 3.360 |
| 2 | E9 | R | Qasrawi 2000 [29] | 0.9490 | 9.1322 |
| 3 | E11 | $V_P$, R | Arioglu et al. [31] | 0.9157 | 10.6093 |
| 4 | E8 | R | Kheder [28] | 0.9491 | 15.5343 |
| 5 | E14 | $V_P$, R | Kheder [34] | 0.9485 | 16.9512 |
| 6 | E10 | $V_P$, R | Logothetis [30] | 0.9139 | 25.5159 |
| 7 | E13 | $V_P$, R | Erdal 2009 [33] | 0.9063 | 28.4509 |
| 8 | E7 | R | Logothetis [30] | 0.9489 | 29.3858 |
| 9 | E1 | $V_P$ | Turgut 2004 [25] | 0.7874 | 34.9199 |
| 10 | E12 | $V_P$, R | Amini et al. [32] | 0.2628 | 38.1740 |
| 11 | E3 | $V_P$ | Trtnik et al. [26] | 0.7819 | 41.7633 |
| 12 | E4 | $V_P$ | I.H.Nash't [27] | 0.7877 | 42.4471 |
| 13 | E6 | $V_P$ | G.F.Kheder [28] | 0.7879 | 45.2446 |
| 14 | E2 | $V_P$ | Logothetis [30] | 0.7870 | 46.4956 |
| 15 | E5 | $V_P$ | G.F.Kheder [28] | 0.7885 | 48.2633 |

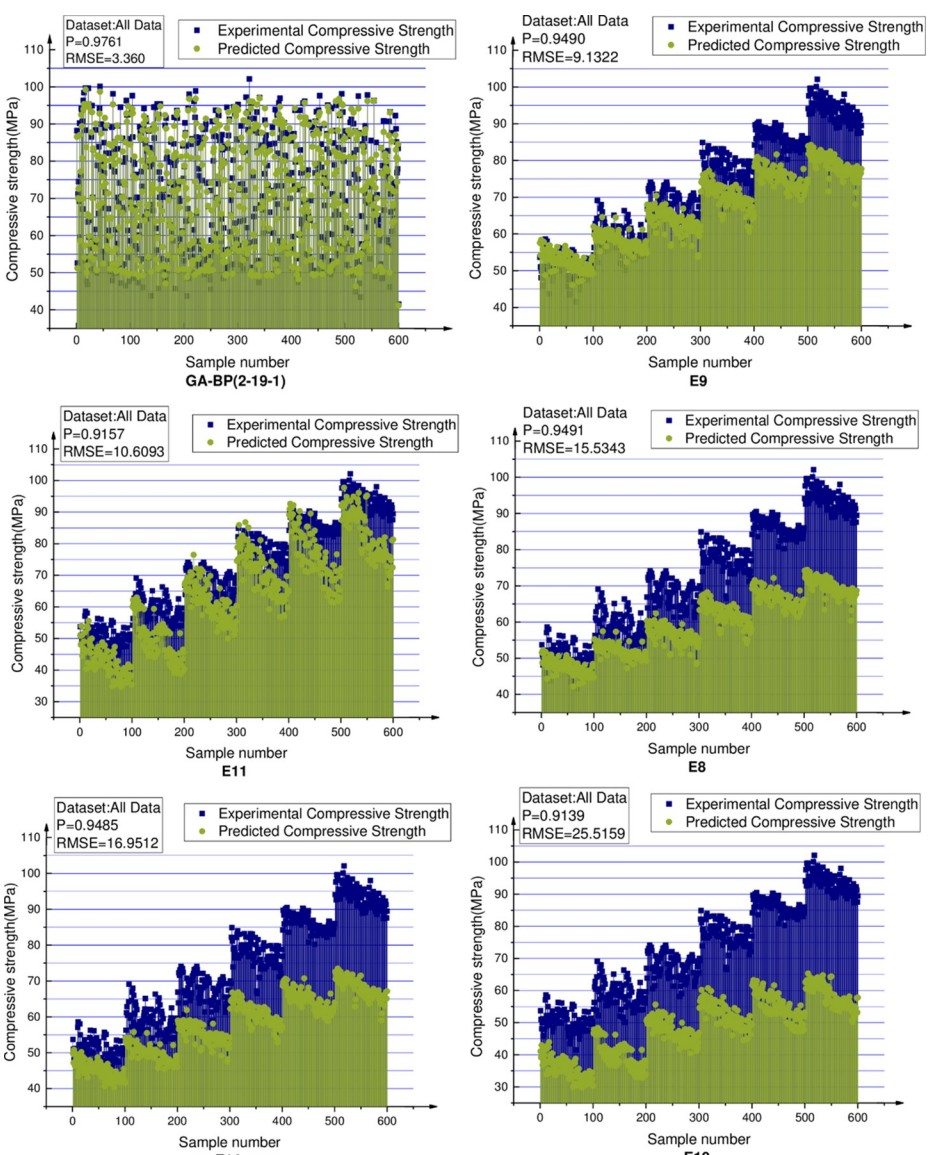

**Fig 25. Comparison of the first six concrete compressive strength prediction models.**

## Conclusion

1. In this study, a BPNN topology consisting of a two-node input layer, a 19-node hidden layer, and a one-node output layer was designed for the NDT of SCC compressive strength. The rebound values and UPVs were experimentally obtained as the dataset and used as input data, which well reflected the SCC strength.

2. To address the problem that the number of neurons in the hidden layer of the BPNN is difficult to determine, the best number of hidden-layer neurons was obtained by using the RMSE as the evaluation index based on 57 tests.

3. In the SCC compressive strength prediction model proposed in this study, the initial weights and thresholds of the traditional BPNN were optimized by GA, which reduced the

proneness of BPNN to be trapped in local extremes. This resulted in higher prediction accuracy than traditional BPNN and increased the correlation coefficient between the test data and prediction results from 0.967 to 0.979, RMSE decreased from 3.703 to 2.972. Therefore, the proposed method can be satisfactorily used for in situ testing of SCC compressive strength. Compared with the traditional method of concrete compressive strength estimation using linear regression equations, the proposed model has relatively high accuracy and produces good results, thereby assisting engineers and researchers in estimating SCC compressive strength.

## Supporting information

**S1 Fig. Comparison of other empirical formulas.**
(PDF)

## Acknowledgments

First, I would like to thank the teachers and senior engineers of the laboratory for providing the experimental scheme for this study. This study provides the experimental basis for the preparation of the special concrete standard.

## Author Contributions

**Conceptualization:** Liangtao Bu, Beixin Lu.

**Data curation:** Guoqiang Du.

**Funding acquisition:** Qi Hou.

**Investigation:** Qi Hou.

**Methodology:** Guoqiang Du.

**Project administration:** Liangtao Bu.

**Resources:** Qi Hou, Jing Zhou.

**Software:** Guoqiang Du.

**Supervision:** Jing Zhou.

**Writing – original draft:** Guoqiang Du.

**Writing – review & editing:** Liangtao Bu.

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
