## [Decision Letter · Decision Letter 0]

16 Feb 2021

PONE-D-21-02136

Prediction of the compressive strength of high-performance self-compacting concrete by an ultrasonic-rebound method based on a GA-BP neural network

PLOS ONE

Dear Dr. Bu,

Thank you for submitting your manuscript to PLOS ONE. After careful consideration, we feel that it has merit but does not fully meet PLOS ONE’s publication criteria as it currently stands. Therefore, we invite you to submit a revised version of the manuscript that addresses the points raised during the review process.

We look forward to receiving your revised manuscript.

Kind regards,

Tianyu Xie, Ph.D.

Academic Editor

PLOS ONE

Journal Requirements:

2.Thank you for stating the following in the Financial Disclosure section:

""Thin layer pull out method for field testing of compressive strength of engineering structure reinforcement" China Construction Industry Design Press, 2017. The preparation of the standard provided funds for this study."

We note that one or more of the authors are employed by a commercial company: Hunan Hongli Civil Engineering Inspection and Testing Co., Ltd.,

Reviewers' comments:

Reviewer's Responses to Questions

**Comments to the Author**

1. Is the manuscript technically sound, and do the data support the conclusions?

Reviewer #1: Yes

Reviewer #2: Yes

2. Has the statistical analysis been performed appropriately and rigorously? 

Reviewer #1: Yes

Reviewer #2: Yes

3. Have the authors made all data underlying the findings in their manuscript fully available?

Reviewer #1: Yes

Reviewer #2: Yes

4. Is the manuscript presented in an intelligible fashion and written in standard English?

Reviewer #1: Yes

Reviewer #2: Yes

5. Review Comments to the Author

Reviewer #1: This paper proposed a GA-BPNN model to predict the compressive strength of SCC to avoid destructive process in in-situ testing. A BPNN model is a well-known neural network method aiming at reducing mean square root through iterative process by minimizing the R value. The author believes the problem of low accuracy and poor robustness of in-situ testing of the compressive strength could be addressed through this new method.

Q1. What is the advantage of the proposed GA-BP method over other neural network models [Ref10-15]? It seems the models in Ref 12 and 15 do better predication in tensile or flexural strength, and the author needs to demonstrate the advantages of GA-BP than others.

Q2. The advantages of the proposed model over the empirical relationships (E1-E14) should be demonstrated as well. A comparative result may be presented in the same figure.

Q3. A typical disadvantage over the empirical relationships is the numerical efficiency, which should be discussed. Suggest the author present the GA-BP training time/ modelling time with a further discussion.

Q4. What is the novelty of this article? Both of the testing method and the BPNN model are well known. The author needs to clearly state the existing research gap and novelty/improvement of the research method.

Q5. It is confusing when the author stating ‘the problem of low accuracy in-situ testing could be addressed through is method’ in the abstract. If the predication is based on in-situ testing and the testing results has low accuracy, how could the author improve the in-situ testing accuracy through this method?

Q6. L064.’To overcome this problem …improve the convergency rate and accuracy of BPNN’. Clarify whether this is the statement by the author or otherwise needs to be referenced.

Q7. L064. The author needs to demonstrate how much efficiency can be improved compare to the other method, e.g., how much modelling time can be saved?

Q8. L205. Suggest rephrase the sentence.

Q9. L251 ‘An appropriate dataset is necessary to train reliable NN model’. Discuss what is an appropriate dataset or otherwise reference the statement.

Q10.L278. Explain how the was best number determined through experiments.

Q11.L279. The sentence ‘the appropriate number of … between 2 and 20’ needs to be referenced.

Q12.L322. If one hidden layer is enough to solve complex engineering problem why the author needs to determine the hidden layer again? E.g., ‘L333. This process involves the determination of the number of hidden layers’.

Q13.L401. ‘Increase the predication results … from 0.928 to 0.939’. and the author claim ‘Therefore this method can be satisfactory used for in situ testing of SCC compressive strength.’ However in engineering practice, overestimating the material strength is often dangerous and it is not appropriate to make the conclusion based on results correlation on R2 only. How could the author control if the model overestimate the strength?

Q14.L401. Does the traditional BPNN and the GP-BP have the same dataset?

Q15.PP39. Fig 4. Needs to update.

Q16.PP47. Fig 12. The orange line needs to be labelled.

Q17. A neural network model’s accuracy is subjected to the size of the training data. The author mentioned a 600 dataset was selected but unclear based on what reason. The author needs to demonstrate the influence of the sample size to the result accuracy. Suggest further parametric analysis.

Reviewer #2: This paper presents a genetic algorithm (GA)-optimized backpropagation neural network (BPNN) model to predict the compressive strength of self-compacting concrete using UPV and rebound value as input parameters. The content is interesting but this reviewer believes there are several critical concerns need to be carried out. Therefore, this reviewer recommends publication of this paper provided that the following major revisions are successfully carried out.

1. The originality of this paper is not clear. The authors must clearly explain what is original in this paper.

2. Please specify the chemical compositions of binders in the mix designs.

3. Section "Model development ANN" contains basically textbook contents on ANN. There is no need to give all the equations from (1) to (7). Therefore, the authors should substantially shorten this section. Please do the same for Section “Performance of the model” and equations (9) to (13).

4. There has to be more substantial discussion in Section “Results and discussion”, not just providing the results.

5. Please use boxplots to show the range of your database in each set of mix designs. You can define x-axis to be the set of your six mix designs (A, B, C, D, E, F) and y-axis to be boxplots for the observed values (for instance fc) in each set of mix designs. Therefore, you have six boxplots in each plot and three plots for fc, Vp and R. Please also compare the standard deviation of each set of 100 test cubes for fc, Vp and R and discuss the reliability of your dataset.

6. R is used for both rebound value and the difference between the predicted and expected values in equations (9) to (13). Please change R for one of them.

7. In lines 279-280, “The appropriate number of hidden-layer neurons is generally between two and 20, and usually there are one or two hidden layers.”. Please give reference.

8. In lines 339-341, “The calculation results are shown in Fig 14. When the number of hidden-layer neurons was 14, the network had the smallest error and the model with the best performance was Model 2-19-1.”

14 or 19?

9. Please run BPNN and GA- BPNN separately for 10 times and show the results (R2, MSE, RMSE, MAE and MAPE) in a table or plot. Then discuss the results.

10. Please discuss the limitations of your method in Conclusion.

11. Please use English language in supplementary file.

6. PLOS authors have the option to publish the peer review history of their article (what does this mean?). If published, this will include your full peer review and any attached files.

Reviewer #1: No

Reviewer #2: No

---

## [Author Response · Author response to Decision Letter 0]

25 Mar 2021

Dear Dr. Tianyu Xie, 

Academic Editor

PLOS ONE

Re: Resubmission of manu reference no. PONE-D-21-02136

On behalf of my co-authors, we thank you very much for giving us an opportunity to revise our manuscript, we appreciate editor and reviewers very much for their positive and constructive comments and suggestions on our manuscript entitled “Prediction of the compressive strength of high-performance self-compacting concrete by an ultrasonic-rebound method based on a GA-BP neural network”. (ID: PONE-D-21-02136).

We also appreciate reviewers for their thought-provoking and constructive comments and suggestions on our manuscript.Those comments are all invaluable and helpful for revising and improving our research. We have carefully studied comments and have made necessary revisions which we hope meet with approval. The responses to reviewer’s comments as well as the corresponding revisions are as follows.

Response to Academic Editor:

1.We have revised the financial disclosure，the updated statement is in the cover letter.

2.All pictures have been adjusted by PACE to ensure that the pictures meet the requirements of plos one.

3.According to your suggestion, we have put the laboratory protocols in protocols.io(DOI: dx.doi.org/10.17504/protocols.io.btjnnkme)

4.The updated Competing Interests Statement is also in the cover letter.

5.The captions of the supporting information file has been attached to the end of the manuscript.

6.All data are available on the publicly accessible database http://doi.org/10.5281/zenodo.4620872

Response to Reviewer 1: Thank you for your review of our paper. We have answered each of your points below.

Q1.[What is the advantage of the proposed GA-BP method over other neural network models [Ref10-15]? It seems the models in Ref 12 and 15 do better predication in tensile or flexural strength, and the author needs to demonstrate the advantages of GA-BP than others.]

Response:The GA-BP model proposed in this study uses GA to optimize the initial weights and thresholds of BPNN, thereby effectively solving the problems of traditional NN models, i.e., slow learning and the susceptibility to being trapped in local extrema. The proposed model determines the optimal NN structure by trial and error. A total of 57 different BPNN models are developed to determine the optimal number of neurons in the hidden layer, resulting in good model prediction performance. In Ref. [12], the field compressive strength of concrete was dynamically predicted based only on the relationship between VP and the concrete compressive strength, which did not produce reliable results, Compared with the dynamic prediction program proposed in Reference [12], the neural network has better prediction performance. The input parameters used in the NN model presented in Ref. [15] were based on the concrete composition (i.e., the cement, fly ash, sand, coarse aggregate and admixture contents and the water-cement ratio), which is usually difficult to obtain for field tests. By contrast, the proposed GA-BPNN model uses the rebound value and sound velocity as input parameters, for which very accurate values can be easily obtained in the field. At the same time, rebound method and ultrasonic method can be used for nondestructive testing

Q2.[ The advantages of the proposed model over the empirical relationships (E1-E14) should be demonstrated as well. A comparative result may be presented in the same figure.]

Response:The proposed NN model has a higher nonlinear fitting ability and prediction accuracy than the traditional empirical relations given by E1 to E14. These empirical relations cannot eliminate the influences of the concrete material and mix proportion on the compressive strength and can only predict the compressive strength for a single material. By contrast, the proposed GA-BPN model can be retrained and applied to concrete materials with different compositions. The predictions obtained using GA-BPNN and the empirical relations are compared in Figure 25 of the revised manuscript with track changes.

Q3.[ A typical disadvantage over the empirical relationships is the numerical efficiency, which should be discussed. Suggest the author present the GA-BP training time/ modelling time with a further discussion]

Response: The numerical efficiency of models is indeed important. NNs have developed from a performance improvement phase to an efficiency enhancement phase. For example, the impressive performance of AlphaGo is obtained by four to six weeks of training on 2000 CPUs and 250 GPUs, with a total power consumption of approximately 600 kW. To increase the low numerical efficiency of NN training, a few acceleration and compression methods are applied to NN models, mainly including network pruning, knowledge distillation, tensor decomposition, transfer learning, parameter quantization, and low-precision NN. The proposed GA-BPNN is not a deep NN and there are a few data in training sets, its numerical efficiency is higher,can be trained in 6 hours on a Home computer. However, a survey of engineering personnel indicates that model accuracy is more important than numerical efficiency for models used in the field. If there are a lot of sample data for training, reaching millions of orders of magnitude, the training speed is often very slow, Because each iteration must perform summation and matrix operations on all samples, the training samples can be divided into several subsets, so that the amount of data contained in each subset is small, and the numerical efficiency of the model will be greatly improved.

Q4.[ What is the novelty of this article? Both of the testing method and the BPNN model are well known. The author needs to clearly state the existing research gap and novelty/improvement of the research method.]

Response:

1. A total of 57 BPNN models are developed in this study based on different input parameters and numbers of neurons in the hidden layer. The optimal NN structure is determined based on the highest model performance.

2. The initial weights and thresholds of traditional BPNN are optimized by GA to satisfactorily address the problems of BPNN, including slow learning and susceptibility to being trapped in local optima.

3. The ultrasonic and rebound methods used in the experiments of this study are both applied to nondestructively test the concrete compressive strength. Only the VP and R values of SCC need to be obtained to predict the SCC compressive strength. These methods can be used to nondestructively test SCC components in the field, which is extremely convenient for field inspectors. 

Q5.[ It is confusing when the author stating ‘the problem of low accuracy in-situ testing could be addressed through is method’ in the abstract. If the predication is based on in-situ testing and the testing results has low accuracy, how could the author improve the in-situ testing accuracy through this method?]

Response:。Methods that are currently used in the field to test concrete compressive strength mainly include the rebound method, the ultrasonic method, the cast-in-place pull-out method, the post-install pull-out method, and the core drilling method. We did not intend to imply that these five test methods are inaccurate. If the above method is used alone, in order to get enough accuracy ,a large amount of sample data needs to be collected, it is difficult to obtain ideal conditions in engineering application. It is the mathematical analysis that many researchers[22-30] use to fit correlation curves for predicting the concrete compressive strength that has a low prediction accuracy. It is based on the ideal conditions in the laboratory, which is different from the conditions in the engineering, so it is difficult to ensure the accuracy of the prediction. The method proposed in this study combines the advantages of the two detection methods,it has loose requirements for application conditions, Can self-improve and learn. Therefore, the method proposed in this study can be used to effectively improve the accuracy and efficiency of field tests for concrete compressive strength.

Q6. [L064.’To overcome this problem …improve the convergency rate and accuracy of BPNN’. Clarify whether this is the statement by the author or otherwise needs to be referenced.]

Response:The respective statement was quoted from Ref. [17], which has been cited in the revised manuscript with track changes.

Q7. [L064. The author needs to demonstrate how much efficiency can be improved compare to the other method, e.g., how much modelling time can be saved?]

Response: NNs have local convergence. If the initial weights are all randomly selected within a local region, the model can easily become trapped in local convergence, significantly decreasing the modelling efficiency. We combine the global search ability of GA with the local search ability of an NN to effectively prevent the NN from being easily trapped in local convergence and to improve the modelling efficiency. However, the extent of the improvement in the efficiency may vary among NN models, and it is difficult to determine the quantity of time saved. The randomness of NNs may also produce different results among runs.

Q8. [L205. Suggest rephrase the sentence.]

Response:We regret the misleading phrasing. The respective sentence has been replaced in the revised manuscript with track changes with “The backpropagation (BP) algorithm is commonly employed to optimize parameters in NN algorithms and BPNN has been widely adopted in civil engineering applications.”

Q9. [L251 ‘An appropriate dataset is necessary to train reliable NN model’. Discuss what is an appropriate dataset or otherwise reference the statement.]

Response:NNs are trained using datasets. Hence, NNs trained using poor datasets naturally perform poorly. A good dataset should have a sufficiently high precision to accurately reflect the relationship between the input and output data. All the experimental data used in this study were obtained under the same experimental conditions, with the same person implementing the test procedure, and with the same test equipment, thereby reducing the measurement error. A good dataset should also encompass all possible cases over as wide a range as possible and contain sufficient data to minimize accidental errors.

Q10.[L278. Explain how the was best number determined through experiments.]

Response: In the present study, the best number of neurons in the hidden layer is determined by testing, that is, by setting the program to try the number of different neurons in the hidden layer one by one, the number of neurons in the hidden layer is determined by the size of the RMSE value of the output results. In view of different input parameters and the number of neurons in the hidden layer, 57 different neural networks are developed, and finally the best neural network structure is determined as per the size of RMSE (2-19-1).

Q11.[L279. The sentence ‘the appropriate number of … between 2 and 20’ needs to be referenced.]

Response: A reference has been added here.

Q12.[L322. If one hidden layer is enough to solve complex engineering problem why the author needs to determine the hidden layer again? E.g., ‘L333. This process involves the determination of the number of hidden layers’.]

Response: Yes, a hidden layer is sufficient to address most prediction issues, which is explained in Reference [42] as well; L333 has been revised here accordingly. Some scholars, such as Joaquín Abellán García 【1】 and Kraiwut Tuntisukrarom 【2】, have determined the number of hidden layers and the number of neurons in the hidden layer through a large number of experiments, which will tremendously decrease the modeling efficiency. Fei Wang et al.【3】 indicated that for networks with few input parameters, one hidden layer is enough, which can address most practical issues. In this paper, there are merely two input parameters. After giving a comprehensive consideration, a hidden layer is determined, and the number of neurons in the hidden layer is determined to be 2-20, and then, the best number of hidden layers is determined by means of experiments. 

【1】Joaquín Abellán García, Jaime Fernández Gómez & Nancy Torres Castellanos(2020): Properties prediction of environmentally friendly ultra-high-performance concrete using artificial neural networks, European Journal of Environmental and Civil Engineering, DOI:10.1080/19648189.2020.1762749

【2】Kraiwut Tuntisukrarom, et al."Prediction of Compressive Strength Behavior of Ground Bottom Ash Concrete by an Artificial Neural Network." Advances in Materials Science and Engineering 2020.(2020):. doi:10.1155/2020/2608231. 

【3】Fei Wang, Zhaofeng Chen, Cao Wu, Yong Yang, Duanyin Zhang & Shun Li(2020): A model for predicting the tensile strength of ultrafine glass fiber felts with mathematics and artificial neural network, The Journal of The Textile Institute, DOI: 10.1080/00405000.2020.1779167

Q13.[L401. ‘Increase the predication results … from 0.928 to 0.939’. and the author claim ‘Therefore this method can be satisfactory used for in situ testing of SCC compressive strength.’ However in engineering practice, overestimating the material strength is often dangerous and it is not appropriate to make the conclusion based on results correlation on R2 only. How could the author control if the model overestimate the strength?]

Response: The performance of the model is evaluated by adding RMSE (root mean square error) to 'Revised Manuscript with Track Changes', which can well exhibit the deviation between the predicted value and the test value, and measure the model more accurately through RMSE and correlation coefficients; the source of errors made in the model is chiefly from data acquisition. The number of measuring points by means of rebound method and ultrasonic method can be increased when data are acquired, so as to reduce the source of errors, which will avoid overestimate the strength of the model. As can be observed from Figure 23 in the revised draft that the prediction deviation of this model can be controlled within the range of 10%, and the only one of the 90 test data has a deviation greater than 10%, which is allowed in the Standard for Inspection and Evaluation of Concrete Strength (GB/T50107-2010).

Q14.[L401. Does the traditional BPNN and the GP-BP have the same dataset?]

Response: In the present study, the same data set is used for the traditional BPNN and GA-BPNN. 

Q15.[PP39. Fig 4. Needs to update.]

Response: Figure 4 has been revised.

Q16.[PP47. Fig 12. The orange line needs to be labelled.]

Response: Figure 12 has been revised.（Fig 14 in the revised version）

Q17. [A neural network model’s accuracy is subjected to the size of the training data. The author mentioned a 600 dataset was selected but unclear based on what reason. The author needs to demonstrate the influence of the sample size to the result accuracy. Suggest further parametric analysis.]

Response: The neural network is trained and learned from the existing data. In the event that the existing data is inaccurate, the learning effect of the model will be decreased. If there are too few training data, data contingency and errors will occur, and the learning effect of neural network will be poor as well. Too much data is uneconomical. So far as it is possible, In this study, six test blocks of different strength grades and 100 test blocks of each strength grade were made, and 600 data points were obtained. For analyzing parameters of this data set, please refer to Table 8(L284) in 'Revised Manuscript with Track Changes'.

Response to Reviewer 2: Thank you for your review of our paper. We have revised the article according to your suggestions.

1. [The originality of this paper is not clear. The authors must clearly explain what is original in this paper.]

Response: In this paper, the author found that it is difficult to detect the compressive strength of SCC in engineering site and the accuracy is low, and then came up with a new prediction model of SCC compressive strength GA-BPNN, which is convenient for engineers to detect the compressive strength of SCC on site. This is not done by predecessors, so this paper is original in a certain sense. Unlike Duan【1】, Anderson, Seal【2】, Aderaw【3】, etc., who have never probed into the research content hereof, of this paper addresses the issues that the data are difficult to obtain in the field and the neural network model is easy to fall into local optimum.

【1】Duan ZH, Kou SC, Poon CS. Prediction of compressive strength of recycled aggregate concrete using artificial neural networks. Constr Build Mater. 2013;40: 1200-1206.

【2】Anderson DA, Seals RK. Pulse velocity as a predictor of 28-and 90-day strength. ACI J Proc. 1981;78: 116-122.

【3】Aderaw, M., Muse, S., & Abiero, Z. C. (2018). Artificial neural network based modelling approach for strength prediction of concrete incorporating agricultural and construction wastes. Construction and Building Materials, 190, 517–525.

2. [Please specify the chemical compositions of binders in the mix designs.]

Response: Chemical composition of binders has been described in Table 1（L109） of 'Revised Manuscript with Track Changes'.

3. [Section "Model development ANN" contains basically textbook contents on ANN. There is no need to give all the equations from (1) to (7). Therefore, the authors should substantially shorten this section. Please do the same for Section “Performance of the model” and equations (9) to (13).]

Response: In the 'Revised Manuscript with Track Changes', the parts of formulas (1)-(7) associated with neural network textbooks have been deleted, while in 'Revised Manuscript with Track Changes', formulas (12) and (13) have been deleted, thus simplifying this part. 

4. [There has to be more substantial discussion in Section “Results and discussion”, not just providing the results.]

Response: In this paper, the results and discussions are discussed in a more substantive way. Parameter setting of neural network is discussed in subsection 'Development of ANN model' of 'Revised Manuscript with Track Changes', and training results of 57 types of neural network structures are displayed in subsection 'Determination of the AN Narchitecture'. Based on RMSE value, the optimal neural network structure is determined. A new subsection 'Comparisions' is added in the revised draft, which compares the proposed neural network model with the empirical relations in the references, better presenting the effect of the proposed GA-BPNN model. 

5. [Please use boxplots to show the range of your database in each set of mix designs. You can define x-axis to be the set of your six mix designs (A, B, C, D, E, F) and y-axis to be boxplots for the observed values (for instance fc) in each set of mix designs. Therefore, you have six boxplots in each plot and three plots for fc, Vp and R. Please also compare the standard deviation of each set of 100 test cubes for fc, Vp and R and discuss the reliability of your dataset.]

Response: The box diagram (Fig. 6-8) is drawn as suggested; the standard deviation of fc,Vp and R of each group of 100 data and the range of data set are listed in Table 8 (L284) of 'Revised Manuscript with Track Changes', and L285-L292 of 'Revised Manuscript with Track Changese' list the reliability of data set. 

6. [R is used for both rebound value and the difference between the predicted and expected values in equations (9) to (13). Please change R for one of them.]

Response: R in formulas (9)-(13) in the Revised Manuscript has been replaced with other characters.

7. [In lines 279-280, “The appropriate number of hidden-layer neurons is generally between two and 20, and usually there are one or two hidden layers.”. Please give reference.]

Response: References [42-47] cited here have been revised. Many scholars A.J. Tenza-Abril, D. McNeish et al. 【1-2】hold that one hidden layer is sufficient to address most prediction issues, and some scholars Rafat Siddique, Joaquín Abellán García et al 【3-4】can indeed determine the number of hidden layers and neurons by doing a large number of experiments, which will tremendously lower the efficiency of model building. Fei Wang et al.【5】 indicated that for networks with few input parameters, one hidden layer is enough, which can address most practical issues. There are only two input parameters in the model built in this paper. By giving a comprehensive consideration, determine a hidden layer, and confirm the number of neurons in the hidden layer to be 2-20, and make clear the optimal number of neurons in the hidden layer by experiments. 

【1】A.J. Tenza-Abril,Y. Villacampa,A.M. Solak,F. Baeza-Brotons. Prediction and sensitivity analysis of compressive strength in segregated lightweight concrete based on artificial neural network using ultrasonic pulse velocity. Construction and Building Materials.2018;189:1173-1183.

【2】D. McNeish, On using Bayesian methods to address small sample problems,Struct. Equ. Modelling (2016), https://doi.org/10.1080/10705511.2016.1186549.

【3】Rafat Siddique,Paratibha Aggarwal,Yogesh Aggarwal. Prediction of compressive strength of self-compacting concrete containing bottom ash using artificial neural network. Advances in Engineering Software.2011;42:780-786.

【4】Joaquín Abellán García, Jaime Fernández Gómez & Nancy Torres Castellanos(2020): Properties prediction of environmentally friendly ultra-high-performance concrete using artificial neural networks, European Journal of Environmental and Civil Engineering, DOI:10.1080/19648189.2020.1762749.

【5】Fei Wang, Zhaofeng Chen, Cao Wu, Yong Yang, Duanyin Zhang & Shun Li(2020): A model for predicting the tensile strength of ultrafine glass fiber felts with mathematics and artificial neural network, The Journal of The Textile Institute, DOI: 10.1080/00405000.2020.1779167

8. [In lines 339-341, “The calculation results are shown in Fig 14. When the number of hidden-layer neurons was 14, the network had the smallest error and the model with the best performance was Model 2-19-1.”14 or 19?]

Response:In the 'Revised Manuscript with Track Changes', the author designed the structure of neural network further, trained 57 BP neural networks of different structures by using MATLAB software, and selected the best structure of neural network as 2-19-1 based on RMSE value.Due to the author's negligence, 14 should be changed to 19. 

9. [Please run BPNN and GA- BPNN separately for 10 times and show the results (R2, MSE, RMSE, MAE and MAPE) in a table or plot. Then discuss the results.]

Response: The running results of BPNN without optimized by genetic algorithm and GA-BPNN model optimized by genetic algorithm have been given in subsection 'Optimization by GA' (L421) of 'Revised Manuscript with Track Changes' (Fig 20, Fig 23), and the prediction results of the two models are elaborated as well. In this study, correlation coefficient (P) and root mean square error (RMSE) were selected to evaluate the performance of the model.

10. [Please discuss the limitations of your method in Conclusion.]

Response: Limitations (L502) are added to the 'Revised Manuscript with Track Changes'.

11. [Please use English language in supplementary file.]

Response: well

We hope the revised version is now suitable for publication and look forward to hearing from you in due course. We look forward to hearing from you regarding our submission. We would be glad to respond to any further questions and comments that you may have. We shall look forward to hearing from you at your earliest convenience.

Sincerely,

liangtao Bu

Professor

School of Civil Engineering, Hunan University

---

## [Decision Letter · Decision Letter 1]

14 Apr 2021

Prediction of the compressive strength of high-performance self-compacting concrete by an ultrasonic-rebound method based on a GA-BP neural network

PONE-D-21-02136R1

Dear Dr. Bu,

We’re pleased to inform you that your manuscript has been judged scientifically suitable for publication and will be formally accepted for publication once it meets all outstanding technical requirements.

Kind regards,

Tianyu Xie, Ph.D.

Academic Editor

PLOS ONE

Reviewers' comments:

Reviewer's Responses to Questions

**Comments to the Author**

1. If the authors have adequately addressed your comments raised in a previous round of review and you feel that this manuscript is now acceptable for publication, you may indicate that here to bypass the “Comments to the Author” section, enter your conflict of interest statement in the “Confidential to Editor” section, and submit your "Accept" recommendation.

Reviewer #1: All comments have been addressed

Reviewer #2: All comments have been addressed

2. Is the manuscript technically sound, and do the data support the conclusions?

Reviewer #1: Yes

Reviewer #2: Yes

3. Has the statistical analysis been performed appropriately and rigorously? 

Reviewer #1: Yes

Reviewer #2: Yes

4. Have the authors made all data underlying the findings in their manuscript fully available?

Reviewer #1: Yes

Reviewer #2: Yes

5. Is the manuscript presented in an intelligible fashion and written in standard English?

Reviewer #1: Yes

Reviewer #2: Yes

6. Review Comments to the Author

Reviewer #1: The author has presentated a detailed discussion and addressed all the reviewer comments hence this paper has meet the minimum publish criteria.

Reviewer #2: (No Response)

7. PLOS authors have the option to publish the peer review history of their article (what does this mean?). If published, this will include your full peer review and any attached files.

Reviewer #1: No

Reviewer #2: No

---

## [Editor Report · Acceptance letter]

19 Apr 2021

PONE-D-21-02136R1 

Prediction of the compressive strength of high-performance self-compacting concrete by an ultrasonic-rebound method based on a GA-BP neural network 

Dear Dr. Bu:

I'm pleased to inform you that your manuscript has been deemed suitable for publication in PLOS ONE. Congratulations! Your manuscript is now with our production department. 

Kind regards, 

on behalf of

Dr. Tianyu Xie 

Academic Editor

PLOS ONE